# Winds and temperatures of the Arctic middle atmosphere during January measured by Doppler lidar

Jens Hildebrand[1], Gerd Baumgarten[1], Jens Fiedler[1], and Franz-Josef Lübken[1]

[1]Leibniz-Institute of Atmospheric Physics at the Rostock University, Kühlungsborn, Germany

*Correspondence to:* G. Baumgarten (baumgarten@iap-kborn.de)

**Abstract.** We present an extensive data set of simultaneous temperature and wind measurements in the Arctic middle atmosphere. It consists of more than $300\,\mathrm{h}$ of Doppler Rayleigh lidar observations obtained during three January seasons 2012, 2014, and 2015, and covers the altitude range from $30\,\mathrm{km}$ up to about $85\,\mathrm{km}$. The data set reveals large year-to-year variations of monthly mean temperatures and winds, which in 2012 are affected by a sudden stratospheric warming. The temporal evolution of winds and temperatures after that warming are studied over a period of two weeks, showing an elevated stratopause and the reformation of the polar vortex. The monthly mean temperatures and winds are compared to data extracted from the Integrated Forecast System of the European Centre for Medium-Range Weather Forecasts (ECMWF) and the Horizontal Wind Model (HWM07). Lidar and ECMWF data show good agreement of mean zonal and meridional winds below $\approx 55\,\mathrm{km}$ altitude, but we also find mean temperature, zonal wind, and meridional wind differences of up to $20\,\mathrm{K}$, $20\,\mathrm{m\,s^{-1}}$, and $5\,\mathrm{m\,s^{-1}}$, respectively. Differences between lidar observations and HWM07 data are up to $30\,\mathrm{m\,s^{-1}}$. From the fluctuations of temperatures and winds within single nights we extract the potential and kinetic gravity wave energy density (GWED) per unit mass. It shows that the kinetic GWED is typically 5 to 10 times larger than the potential GWED, the total GWED increases with altitude with a scale height of $\approx 16\,\mathrm{km}$. Since temporal fluctuations of winds and temperatures are underestimated in ECMWF, the total GWED is underestimated as well by a factor of 3 to 10 above $50\,\mathrm{km}$ altitude. Similarly, we estimate the energy density per unit mass for large-scale waves (LWED) from the fluctuations of nightly mean temperatures and winds. The total LWED is roughly constant with altitude. The ratio of kinetic to potential LWED varies with altitude over two orders of magnitude. LWEDs from ECMWF data show results similar as the lidar data. From the comparison of GWED and LWED, it follows that large-scale waves carry about 2 to 5 times more energy than gravity waves.

# 1 Introduction

Winds in the middle atmosphere play an important role for atmospheric dynamics; e.g., filtering of gravity waves is controlled by the background wind field (e.g., Lindzen, 1981; Gill, 1982; Nappo, 2002). As these gravity waves transport energy and momentum over long distances, winds indirectly affect large-scale circulations (e.g., Geller, 1983; Holton, 1983). Therefore, wind measurements in the middle atmosphere with reasonable temporal and vertical resolution are of special interest (Meriwether and Gerrard, 2004; Drob et al., 2008). Not only do wind measurements provide additional information about atmospheric stability, together with temperature observations they also offer more sophisticated studies of gravity waves (e.g., Eckermann et al., 1995; Zink and Vincent, 2001; Placke et al., 2013; Bossert et al., 2014; Baumgarten et al., 2015) than studying gravity waves solely from temperature measurements (e.g., Chanin and Hauchecorne, 1981; Whiteway and Carswell, 1995; Alexander et al., 2011). In a recent study, Dörnbrack et al. (2017) point out that information about background wind is essential to correctly interpret ground-based gravity wave observations, specifically regarding identified phase lines and the vertical propagation direction. However, simultaneous wind and temperature measurements covering a wider altitude range of the middle atmosphere are rare (e.g., Goldberg et al., 2004). The main reason is the technical challenge of wind measurements in these altitudes. MST and MF radars do not cover the altitude range between 20 and 60 km due to the absence of free electrons; whereas the altitude range of meteor radars starts at $\approx 80$ km altitude (see, e.g., Fig. 1 in Baumgarten, 2010). Balloons reach only top altitudes of 30–40 km. Meteorological rockets, equipped with chaff, falling spheres or starutes, are able to measure winds in the entire middle atmosphere between about 20 and 100 km (e.g., Widdel, 1987, 1990; Schmidlin et al., 1991; Lübken and Müllemann, 2003; Müllemann and Lübken, 2005). Such rocket soundings yield a reasonable vertical resolution, but are conducted only sporadically. Data from several campaigns at Arctic sites, which cover longer periods, have been published by, e.g., Meyer et al. (1987), Lübken and Müllemann (2003), and Müllemann and Lübken (2005). Microwave radiation is used to measure the Doppler shift of thermally excited molecules. This technique is used, e.g., by the MLS instrument onboard the Aura satellite (Wu et al., 2008) and the ground-based WIRA instrument (Rüfenacht et al., 2012, 2014), and had been used by the SMILES instrument onboard the ISS (Baron et al., 2013). Another approach is to measure the Doppler shift of airglow lines. This was done by the instruments HRDI and WINDII onboard UARS (Hays et al., 1993; Shepherd et al., 1993); TIDI onboard the TIMED satellite (Killeen et al., 2006) still employs this technique. A ground-based instrument which measures wind speeds by analyzing airglow is ERWIN II (Kristoffersen et al., 2013); since it relies on three dedicated airglow emissions only, its height range is limited to layers between 87 and 97 km altitude. An indirect approach to estimate wind speeds from satellite observations is to retrieve geostrophic winds from geopotential heights on fixed pressure levels (e.g., Randel, 1987). The lidar technique allows to derive wind speeds directly from measuring the Doppler shift of light backscattered at moving particles. Resolving the Doppler shift is technically challenging and wind lidars are therefore sophisticated instruments. While sodium resonance lidars yield wind speeds in the sodium layer between about 80 km and 105 km altitude (e.g., Liu et al., 2002; She et al., 2002; Franke et al., 2005; Yuan et al., 2012), Rayleigh lidars cover mainly altitudes below 50 km (e.g., Tepley, 1994; Friedman et al., 1997; Souprayen et al., 1999; Huang et al., 2009; Xia et al., 2012). Reports about regular wind measurements by lidar are scarce: Tepley (1994) presents winds between 10 and 60 km altitude, derived during 43 nights at the tropical site

Arecibo; Souprayen et al. (1999) derived horizontal winds during 170 nights in the altitude range 8–50 km at mid latitudes; regular observations of horizontal winds with sodium resonance lidars (80–105 km) were presented by Franke et al. (2005) and Yuan et al. (2012) for tropical and mid latitudes, respectively.

The ALOMAR Rayleigh/Mie/Raman (RMR) lidar is the only instrument that derives both horizontal wind components and temperature simultaneously from the upper stratosphere up to the mesosphere. In this study, we present horizontal winds and temperatures obtained by DoRIS, the Doppler Rayleigh Iodine Spectrometer of the ALOMAR RMR lidar, during the three January seasons 2012, 2014, and 2015, in total more than 300 h of observations. They provide the most extensive data set of simultaneous wind and temperature measurements in the middle atmosphere, and allow us to study the interannual variability of temperatures and winds, the temporal evolution on time scales of days, e.g., after the stratospheric warming in January 2012, and during single nights. This study also analyzes the representation of temperatures and winds by the Integrated Forecast System (IFS) of the European Centre for Medium-Range Weather Forecasts (ECMWF) and the Horizontal Wind Model (HWM07) regarding the comparison to observational data. Subsequently, potential and kinetic energy densities of gravity waves and large-scale waves are calculated and analyzed.

## 2 Instrument

The ALOMAR RMR lidar (69.3°N, 16.0°E) is a twin lidar with two identical transmitting lasers, two identical receiving telescopes and one detection system. It measures temperatures and aerosols in the middle atmosphere on routine basis since 1997 (von Zahn et al., 2000; Schöch et al., 2008). Since 2009 the lidar measures wind speeds as well, using the Doppler Rayleigh Iodine Spectrometer DoRIS (Baumgarten, 2010). Detailed descriptions of the instrumental setup and the wind retrieval as well as initial results for the altitude range 30–85 km were presented by Baumgarten (2010), Hildebrand et al. (2012), and Lübken et al. (2016). Basically, the wind retrieval relies on measuring the Doppler shift of the backscattered light using iodine absorption spectroscopy; temperatures are retrieved by hydrostatic integration of altitude profiles of relative air density (Kent and Wright, 1970; Hauchecorne and Chanin, 1980). The two individually derived temperature profiles for both lasers/telescopes are averaged to one temperature profile; this reduces the measurement uncertainty, but the amplitudes of gravity waves are not affected significantly (since the distance of both sounding volumes is much shorter than typical horizontal wavelengths of the inertia gravity waves which are most prominent in the 1 h averaged profiles: 40 km distance at 80 km altitude compared to wavelengths of several hundred kilometers (e.g., Baumgarten et al., 2015)).

## 3 Data and processing

### 3.1 Data

The data set used for this study was acquired during nights in January 2012, 2014, and 2015. January 2013 is excluded since there exist only about 10 h of nighttime horizontal wind observations. The data were integrated over 1 h. The vertical resolution is 150 m, but data were smoothed with a running window with a size of 3 km. Typical uncertainties are 0.5 K and 3 m s$^{-1}$

at $50\,\mathrm{km}$ altitude but increase with altitude (due to less received backscattered light from higher altitudes, mainly due to decreasing air density). The retrieved temperature and wind speed profiles considered in this study are limited to measurement uncertainties of $\Delta T \leq 5\,\mathrm{K}$ and $\Delta u = \Delta v \leq 20\,\mathrm{m\,s^{-1}}$, respectively. Due to technical issues the lower altitude limit in January 2014 and January 2015 is about $40\,\mathrm{km}$ instead of $30\,\mathrm{km}$. As lidar operations depend on weather conditions, the observations are unequally distributed over the years: $65\,\mathrm{h}$ during seven nights between 19 and 30 January 2012, $170\,\mathrm{h}$ during 16 nights between 10 and 31 January 2014, and $78\,\mathrm{h}$ during five nights between 19 and 24 January 2015. Table 1 lists the nights and the respective duration of the lidar observations. Note that although the sampling is quite sparse in January 2012 and 2015, these are the only available simultaneous wind and temperature observations in the Arctic stratosphere and mesosphere. For the analysis of wave phenomena in Sect.s 4.4 and 4.5 we restrict the data set to nights with observations of at least $10\,\mathrm{h}$; this reduces the number of observations taken into account to two thirds of the entire data set, but the fraction of data taken into account is reduced by only one tenth. Table 2 gives an overview of the observations taken into account for analyses based on all nights and long observations only.

Additionally, model data are used for the location of ALOMAR: The European Centre for Medium-Range Weather Forecasts provides the Integrated Forecast System IFS. We extracted data with horizontal resolution T1279 at the location $69.28°\,\mathrm{N}$, $16.01°\,\mathrm{E}$ (the data are available with horizontal resolution of $0.25°$, we interpolated these horizontally on pressure levels to our location). We use data from the forecast system with a temporal resolution of $1\,\mathrm{h}$; hence, lidar data and ECMWF data have the same temporal sampling. Profiles between midnight and noon were taken from the model run initialized at $00\,\mathrm{UTC}$, profiles between noon and midnight were taken from the $12\,\mathrm{UTC}$ run. For January 2012 we used cycle Cy37r3, and for January 2014 and 2015 we used cycle Cy40r1. Both cycles differ, amongst others, in their vertical resolution, especially at higher altitudes: Cy37r3 has 91 model levels, Cy40r1 has 137 model levels. For each single $1\,\mathrm{h}$ profile the pressure coordinate is converted into geometric altitude; the profile is then interpolated to the vertical resolution of the lidar data. The Horizontal Wind Model HWM07 is an empirical model that accumulates data from different instruments obtained over fifty years (Drob et al., 2008). Therefore, the model does not contain any year-to-year variation, but has more character of a climatology. We extracted data on an hourly basis (corresponding to the temporal sampling of the lidar) for the location $69.3°\,\mathrm{N}$, $16.0°\,\mathrm{E}$.

## 3.2 Gravity wave energy density

We used the following equations (e.g., Geller and Gong, 2010) to derive potential and kinetic gravity wave energy density (GWED) per unit mass from temperature and wind speed fluctuations ($T'$, $u'$, and $v'$, respectively):

$$E_{\mathrm{pot}} = \frac{1}{2}\frac{g^2}{N^2}\left(\frac{T'}{\bar{T}}\right)^2 \qquad\text{and}\qquad E_{\mathrm{kin}} = \frac{1}{2}\left(u'^2 + v'^2\right), \quad (1)$$

with $g$ as gravitational acceleration, $N$ as Brunt–Väisälä frequency, and $\bar{T}$ as background temperature. The fluctuations are derived by subtracting the respective nightly mean profile. As stated by Ehard et al. (2015), applying this method might include tidal signatures in the resulting gravity wave energy densities; furthermore, the resolved GW spectrum depends on the length of an observation, which hinders comparison of GWEDs. Although Ehard et al. (2015) proposed applying a Butterworth filter to extract GWs, we use the nightly mean method since we tested different approaches for background estimation with our lidar

data and found no significant differences in the resulting GWEDs. To accommodate the mentioned drawbacks of the nightly mean method, we apply the following procedure: We take only measurements with at least $10\,\mathrm{h}$ duration into account (since the nightly mean profiles of shorter measurements would include wave-like features); within one night we then select the first ten $1\,\mathrm{h}$ profiles to calculate GWEDs for this time span (therefore, the covered GW spectrum is relatively wide and constant for all observations, although it might contain some short-scale tidal components); we shift the $10\,\mathrm{h}$ window by $1\,\mathrm{h}$ and repeat the GWED calculation as often as the window fits into the observation period of that night (therefore, different phases of possibly included tides are sampled); finally we calculate the mean and the standard deviation of all the GWED profiles of one night (therefore, we can estimate the GWED variability during single nights).

## 4  Results

### 4.1  January variability

For a first descriptive presentation of the data set, Fig. 1 shows mean altitude profiles of temperatures and horizontal winds for Januaries 2012, 2014, and 2015. It is evident that the mean profiles for the three years differ remarkably. While in 2012 highest temperatures of $245\,\mathrm{K}$ occur at $38\,\mathrm{km}$ altitude, highest temperatures in 2014 are $270\,\mathrm{K}$ and occur at $50\,\mathrm{km}$ altitude; the temperatures in 2012 and 2015 show enhanced variability around 70 and $60\,\mathrm{km}$ altitude, respectively, but there is no such enhanced variability in 2014. The strength of the eastward zonal winds varies, too: In 2014 and 2015 highest wind speeds of $50$–$70\,\mathrm{m\,s^{-1}}$ occur around $45\,\mathrm{km}$ altitude, while in 2012 the zonal wind is weak at this height, and the highest zonal wind speeds occur between 62 and $72\,\mathrm{km}$, with enhanced variability. Mean meridional winds even have different directions in different years: In 2012 it is mainly northward, in 2014 it has no predominant direction, and in 2015 it is mainly southward.

Besides this noticeable year-to-year variations we find large variability within the Januaries of the different years. The standard deviations of temperature data at $50\,\mathrm{km}$ and $70\,\mathrm{km}$ altitude are 6 K and 21 K in January 2012, 8 K and 7 K in January 2014, and 4 K and 9 K in January 2015; noteworthy is the increased standard deviation of $18\,\mathrm{K}$ at $60\,\mathrm{km}$ altitude in January 2015. The standard deviations of zonal and meridional wind data are of nearly same size ($\pm 2\,\mathrm{m\,s^{-1}}$), namely at $50\,\mathrm{km}$ respectively $70\,\mathrm{km}$ altitude: $18\,\mathrm{m\,s^{-1}}$ and $29\,\mathrm{m\,s^{-1}}$ in January 2012, $24\,\mathrm{m\,s^{-1}}$ and $26\,\mathrm{m\,s^{-1}}$ in January 2014, and $20\,\mathrm{m\,s^{-1}}$ and $30\,\mathrm{m\,s^{-1}}$ in January 2015.

Concluding from the remarkable year-to-year variations and variabilities within Januaries of different years: The polar middle atmosphere in January cannot be described by one single "winter state", and it is not appropriate to infer a general statement or even a climatology from observations of only a few seasons. To investigate the variations in one single month an example is shown in the next section.

### 4.2  Elevated stratopause and polar-vortex reformation after minor SSW in January 2012

During winters, variability in the polar middle atmosphere is mainly caused by planetary waves and sudden stratospheric warmings (SSW): Depending on their type and strength, the polar vortex may be weakened, displaced, or even split; warmer

air from mid-latitudes may intrude into the polar region (e.g., Matsuno, 1971; Labitzke, 1972). The number of SSWs during one season and the time at which they appear vary from year to year (e.g., Labitzke and Kunze, 2012). Around 15 January 2012 a minor SSW, which was a vortex displacement event, occurred (Chandran et al., 2013; Matthias et al., 2013). The ALOMAR RMR lidar took data during the following days and weeks, i.e., in the aftermath of the SSW. Figure 2 shows the temporal

evolution of temperature and zonal and meridional wind after the SSW, starting on 19 January until 4 February. Except for the double-stratopause structure, the temperature profiles from 19 January do not look unusual; the temperature increase between 70 and 80 km altitude indicates a mesospheric inversion layer, whose investigation is, however, beyond the scope of this study. In contrast, the westward zonal winds are exceptional for winter, which is probably a result of the vortex displacement. The strength and relative position of the polar vortex can be inferred from the potential vorticity: Rex et al. (1998) define

36 PVU at the 475 K potential temperature level as the edge of the polar vortex. Based on this definition and using potential vorticity and potential temperature from ECMWF data, we find that ALOMAR is situated inside the polar vortex during that night. It has to be kept in mind that the polar vortex might bend and twist and therefore the vortex location as defined at 475 K ($\approx 19$ km altitude) may not always represent the situation in the upper strato- and mesosphere. Only a few days later (21/22 and 22/23 January) the stratopause was $\approx 15$ to 20 K colder and the upper mesosphere around 70 km altitude was

$\approx 15$ to 20 K warmer; zonal winds were weakly eastward over the entire altitude range and meridional winds developed from weakly southward toward weakly northward with only small variations in altitude. In the first of these two nights the polar vortex edge was above ALOMAR, while in the second night ALOMAR was situated outside the vortex. Baumgarten et al. (2015) show time-altitude sections of temperature and wind data of this period, which exhibit very pronounced gravity wave structures. During the following week, the thermal and dynamic structure over ALOMAR changed remarkably: On 28/29

and 29/30 January, the temperature maximum around 40 km altitude vanished and the highest temperatures shifted upward to around 70 km altitude; at roughly the same altitude where maxima of zonal and meridional wind occurred. ALOMAR was again situated inside the polar vortex. During the beginning of February, the maxima in temperature, zonal wind, and meridional wind intensified and descended further. These phenomena are closely connected to the preceding SSW: They are referred to as elevated stratopause and reformation of the polar vortex, which sometimes occur after stratospheric warmings (e.g., Labitzke,

1972; Manney et al., 2009). In contrast to this work, those two studies analyzed vortex split events with a complete breakdown of the polar vortex.

Concluding, the minor SSW of 2012 is peculiar: It is followed by an elevated stratopause event, although it is neither a major warming nor a vortex split event. Thus, this observation is evidence that elevated stratopause events can occur even after minor SSW, as previously stated by de la Torre et al. (2012) and Chandran et al. (2013). Although the basic mechanisms of elevated

stratopauses and the polar vortex reformation are known (e.g., Tomikawa et al., 2012) and temperatures and zonal mean zonal winds were derived previously (winds only indirectly from geopotential-height observations by satellites (e.g., Manney et al., 2009)), this is the first time to our knowledge that an elevated stratopause together with the reformation of the polar vortex have been observed with a direct temperature and wind measurement technique. These unique observations reveal features which are not represented in ECMWF data, which highlights the need for observations of such peculiar events to broaden the

data basis against which models can be compared to test their fidelity. The differences, which are present in temperature and

wind data as well, highlight the importance of local observations with adequate spatial and temporal resolution, and will be discussed in detail in the following section.

## 4.3 Comparison to models

Figure 2 includes data extracted from ECMWF. Especially above $50\,\mathrm{km}$ altitude the comparison between lidar and ECMWF is dissatisfying, particularly for the end of January and beginning of February: The elevated stratopause and the reformation of the polar vortex are not captured in ECMWF. This yields to differences of up to $40\,\mathrm{K}$ and $20\,\mathrm{m\,s^{-1}}$, respectively. One explanation for the poor comparison might be that this period was affected by an SSW. Therefore, we compare lidar data with ECMWF and HWM07 data for the whole data set, which is shown in Fig. 3: It depicts the same lidar profiles as Fig. 1 and mean profiles taken from ECMWF for January 2012 (panel a), January 2014 (b), and January 2015 (c), and data cumulated over all three seasons, including HWM07 (d). Note that all three data sets have the same temporal sampling. The standard deviation is calculated as the deviation of all $1\,\mathrm{h}$ profiles of one month from the monthly mean profile, which is calculated from these $1\,\mathrm{h}$ profiles.

We first concentrate on HWM07 data (panel d, winds only). Although HWM07 is more like a climatology without any year-to-year variation, some studies use it as representation of mean or background wind fields, even for single case studies, (e.g., Assink et al., 2012; Hedlin and Walker, 2012; Fee et al., 2013). However, HWM07 describes the actual winds inadequately: The zonal wind is too weak in the upper stratosphere (compared to ECMWF) and too strong in the upper mesosphere (compared to lidar), with differences are up to $20\,\mathrm{m\,s^{-1}}$; in between mean zonal wind matches quite well. HWM07's meridional wind is northward in the entire altitude range, while the mean observed meridional wind is weakly southward below $60\,\mathrm{km}$ altitude and weakly northward above; differences are on the order of $30\,\mathrm{m\,s^{-1}}$. It is remarkable that the meridional wind in HWM07 is not only of different magnitude than the observed winds but also of different direction. The temporal variability (indicated by the standard deviation) is much smaller than for the lidar data. One reason for this discrepancy, aside from the missing year-to-year variations in HWM07, is the limited number of observations taken into account in HWM07 for this location and altitude range (see Tab. 1 in Drob et al. (2008)).

Comparison with ECMWF data: The data of 2014 and 2015 were not affected by SSWs, but still the temperature comparison between lidar and ECMWF is not good: The stratopause is too cold (up to $10\,\mathrm{K}$) and too low (up to $4\,\mathrm{km}$) in ECMWF; at higher altitudes temperatures from ECMWF are much too low, namely up to $25\,\mathrm{K}$. This can also be seen in panel (a) of Fig. 4, which shows altitude profiles of the mean of the hourly differences ($\Delta x = \frac{1}{N}\sum(x_{\mathrm{ECMWF}} - x_{\mathrm{lidar}})$), including the respective standard deviation and the standard error of the mean for the lidar data. Regarding zonal winds, the comparison between ECMWF and lidar is nonuniform for the three years: In 2012 and 2014 it is very good below $60\,\mathrm{km}$ altitude with mean differences of $2\,\mathrm{m\,s^{-1}}$ or less, while above $60\,\mathrm{km}$ altitude mean differences are up to $20\,\mathrm{m\,s^{-1}}$ and $15\,\mathrm{m\,s^{-1}}$, respectively; in 2015 mean differences between $10$ and $20\,\mathrm{m\,s^{-1}}$ occur throughout the altitude range of $45$ to $70\,\mathrm{km}$. For meridional winds the comparison is much better: Mean differences are mostly smaller than or around $5\,\mathrm{m\,s^{-1}}$ only, hence on the same order as the standard error of the mean of the lidar data. Similar results concerning ECMWF temperatures in the middle and upper mesosphere were reported by, e.g., Le Pichon et al. (2015). They state that the wave-like pattern of the difference profile might be caused by a quasi-stationary planetary wave structure. A study by Rüfenacht et al. (2014) applying wind radiometry found good agreement of

observed winds and ECMWF wind data in the stratosphere, but deviations in the mesosphere up to 50% of the true wind speeds. Please note that the ECMWF IFS cycles used in these studies differ from the ones used in this study.

Figure 4(b) shows distributions of differences between ECMWF and lidar on an hourly basis for different altitude ranges. The distributions of differences are broader for higher altitudes; some distributions are not symmetrical, indicating systematic under- or overestimations for the respective measure. This is especially true for temperatures and zonal winds above 50 km altitude; but does not appear for meridional winds in the entire altitude range covered.

This leads to studying the comparison of lidar and ECMWF data on shorter time scales: Figure 5 shows all 1 h profiles of temperature, zonal, and meridional wind speed, derived by lidar during the night 20/21 January 2015 (between 14:40 UTC and 07:30 UTC) and extracted from ECMWF corresponding to the temporal and altitude sampling of the lidar. Despite the differences between the mean lidar and ECMWF profiles, it is obvious that the lidar data show a larger variability in altitude and time. These differences on smaller scales are the reason for the width of the distribution of differences shown in Fig. 4(b). Despite the differences of single 1 h profiles or nightly mean profiles in principle, the smaller temporal and vertical variability in ECMWF data might indicate that the amount of energy and momentum which is transported by waves is underestimated in ECMWF, which might cause part of the discrepancies of the mean state as shown in Fig. 4(a).

To study the comparison of the variability of each data set in more detail, the dashed lines in Fig. 5 show the root mean square (RMS) of the fluctuations of the 1 h profiles, hence their variability. The RMS of the lidar data increases with altitude, indicating an increase of the amplitudes of the temperature and wind fluctuations (note that the RMS increases faster and is always larger than the mean measurement uncertainty of the lidar data). This is what is expected for the effect of gravity waves, as their amplitudes increase with altitude due to the decreasing air density. In contrast, the RMS profiles of the ECMWF data do not show a general increase with altitude and in large part of the altitude range the RMS of the ECMWF data is smaller than the RMS of the lidar data. This is also true for the whole data set, as can be seen in Fig. 4(c): For each night with at least ten hours of data the RMS of the lidar data and the RMS of the ECMWF data are calculated. Then the monthly average of the ratio of both is calculated and drawn. In general, the higher in altitude the worse is the actual variability represented in ECMWF. Above $\approx 75$ km altitude the ECMWF variability is only one tenth of the variability observed by lidar; one exception is the temperature in January 2012, when the ECMWF variability even at high altitudes is about one third of the lidar variability. Similar results regarding the height-dependent underestimation of gravity wave amplitudes were also reported by Schroeder et al. (2009). From a comparison of model data with global satellite observations they infer that temperature amplitudes in ECMWF are underestimated by a factor of 2 at 28 km altitude and more than five times above 40 km altitude. The reason for the underestimation of the variability at higher altitudes are likely damping mechanisms that are applied in the ECMWF model; an extensive overview of several such approaches is given by Jablonowski and Williamson (2011).

Concluding, ECMWF and especially HWM07 do not represent the thermal and dynamic state of the middle atmosphere sufficiently, neither regarding January mean profiles nor the variability within individual nights, which are underestimated in ECMWF data. This distinct underestimation of the temporal variability of temperatures and winds affects the calculated energy budget of gravity waves, which are the main source of fluctuations on the scale of a few hours. Resulting gravity wave energy densities are discussed in the next section.

## 4.4 Gravity wave energy density

The combination of simultaneous wind and temperature measurements allows us to perform wave studies in more detail. For instance, the energy budget of gravity waves consists of potential and kinetic gravity wave energy. While the former depends on the temperature fluctuations, the latter is based on the wind speed fluctuations. As an example, the left panel of Fig. 6 shows vertical profiles of potential and kinetic GWED for the night 20/21 January 2015. Except at around $47\,\mathrm{km}$ and $52\,\mathrm{km}$ altitude, the kinetic GWED is larger than the potential GWED, mostly by four to five times (shown in the right panel of Fig. 6). As expected from Eq. (1) the potential GWED shows minima and maxima at the same altitudes as the minima and maxima of the temperature fluctuations (cf. Fig. 5); while the kinetic GWED correlates to features of zonal and meridional wind fluctuations (e.g., the minimum of kinetic GWED at $67\,\mathrm{km}$ altitude).

The middle panel of Fig. 6 shows the total GWED. Between $47$ and $53\,\mathrm{km}$ altitude, and above $67\,\mathrm{km}$ the total GWED increases with altitude. In between is a layer of nearly constant total GWED where the kinetic GWED is roughly constant and the potential GWED slightly decreases. A possible reason might be the near adiabatic temperature gradient between $50$ and $60\,\mathrm{km}$ altitude (some profiles show gradients of $\approx -7\,\mathrm{K\,km^{-1}}$), which hinders the upward propagation of gravity waves.

The right panel of Fig. 6 shows the ratio of kinetic to potential GWED and the intrinsic period $2\pi\hat{\omega}^{-1}$ that a monochromatic low- or medium-frequency gravity wave with the given $E_{\mathrm{pot}}$ and $E_{\mathrm{kin}}$ would have (Geller and Gong, 2010):

$$\hat{\omega} = \pm f\sqrt{\frac{E_{\mathrm{kin}}/E_{\mathrm{pot}}+1}{E_{\mathrm{kin}}/E_{\mathrm{pot}}-1}}\,, \tag{2}$$

with the Coriolis parameter $f = 2\Omega\sin\phi$ ($\Omega$: angular speed of Earth's rotation, $\phi$: latitude of observation). We have shown earlier that at times of quasi-monochromatic waves the intrinsic periods calculated from the energy ratios agree to the results of the hodograph method (Baumgarten et al., 2015). While the hodograph method can only be applied in the case of a quasi-monochromatic wave – because it would otherwise be hard or even impossible to identify an ellipse from the zonal and meridional wind fluctuations –, the energy ratio method is applicable also to wind and temperature fluctuations caused by various waves, keeping in mind that the derived $2\pi\hat{\omega}^{-1}$ is not the intrinsic period of a certain wave. However, the method has been applied previously to data sets probably affected by superposition of various gravity waves (e.g., Geller and Gong, 2010; Baumgarten et al., 2015). Note that since temperature and horizontal wind fluctuations are more sensitive to long-period gravity waves than to short-period gravity waves, the energy ratio method is biased toward long-period gravity waves, as stated by Lane et al. (2003) and evaluated by Geller and Gong (2010, their App. A). Nevertheless, due to the temporal integration of the data presented here, short-period gravity waves are discarded anyway. The retrieved $2\pi\hat{\omega}^{-1}$ is larger than $8\,\mathrm{h}$ in most parts; highest values are about $11\,\mathrm{h}$, reasonably smaller than the upper limit of $2\pi f^{-1} = 12.82\,\mathrm{h}$. According to the relationship for the group velocity vector (e.g., Fritts and Alexander, 2003)

$$(c_{\mathrm{g}x}, c_{\mathrm{g}y}, c_{\mathrm{g}z}) = (\bar{u}, \bar{v}, 0) + \frac{[k(N^2-\hat{\omega}^2), l(N^2-\hat{\omega}^2), -m(\hat{\omega}^2-f^2)]}{\hat{\omega}\left(k^2+l^2+m^2+\frac{1}{4H^2}\right)}, \tag{3}$$

with $k$, $l$, $m$ as zonal, meridional, and vertical wave number, respectively, this indicates a more horizontal wave propagation, as $\hat{\omega}^2 - f^2 \to 0$ (and $\hat{\omega}^2 \ll N^2$). The two pronounced minima of $2\pi\hat{\omega}^{-1}$ around $46\,\mathrm{km}$ and $53\,\mathrm{km}$ altitude are caused by

equality of potential and kinetic GWED; wind fluctuations are quite low at these altitudes, while the temperature fluctuations are quite large. This then indicates waves which propagate more vertically, as the weight of $N^2 - \hat{\omega}^2$ in Eq. (3) decreases and the weight of $\hat{\omega}^2 - f^2$ increases. The different vertical-to-horizontal propagation conditions at $46\,\text{km}$ and $53\,\text{km}$ compared to the remaining altitude ranges may have different causes: 1. different origin of the waves; 2. changing background propagation

conditions, i.e., filtering/Doppler shift due to the strong zonal wind shear at these altitudes, reducing wind speeds from $80\,\text{m s}^{-1}$ to $20\,\text{m s}^{-1}$. A clear distinction between these possible explanations is not possible: While the second option is clearly visible in Fig. 5 (large temperature gradient and strong wind shear), the first option can not be excluded. However, a detailed investigation of propagation conditions is beyond the scope of this study.

Figure 6 includes also GWEDs and the $2\pi\hat{\omega}^{-1}$ derived from ECMWF data for the same time period. In the lower part (up to

$\approx 50\,\text{km}$ altitude), the GWEDs are comparable to the lidar data. Above, the total GWED derived from ECMWF data decreases with altitude. Therefore, at $70\,\text{km}$ altitude the GWEDs derived from ECMWF data are nearly two orders of magnitude too small. The kinetic-to-potential GWED ratio is on the same order as the GWED ratio derived by lidar, although the shapes differ, yielding differing profiles of $2\pi\hat{\omega}^{-1}$.

Are these results special or typical? Figure 7 shows mean GWEDs for January 2012, 2014, and 2015, derived from li-

dar (panel a) and ECMWF data (panel b). For this, altitude profiles of GWED of all nights with at least $10\,\text{h}$ of data were averaged. Comparing Figs. 6 and 7(a), the data from 20/21 January 2015 is not unusual. Although the mean total GWED of January 2015 increases nearly throughout the altitude range (in contrast to the data of 20/21 January 2015), the increase is slightly steeper below $\approx 55\,\text{km}$ altitude than it is above. The same is true for January 2014. In January 2012 the GWED between $40$ and $60\,\text{km}$ altitude is somewhat smaller than in January 2014 and 2015. The increase of total GWED with altitude

exhibits a scale height of $\approx 16\,\text{km}$. This is 2.3 times larger than the pressure scale height of $7\,\text{km}$; a relation previously obtained by Fritts and VanZandt (1993) by posing a model gravity wave spectrum. The same scale height was found by Kaifler et al. (2015), although they observed potential energy densities only. Similar scale heights for total energy density and potential energy density would imply a kinetic-to-potential GWED ratio constant with altitude. However, our observations show that the kinetic-to-potential GWED ratio is typically between $5$ and $10$ and slightly increases with altitude, as can be seen in the right

panel of Fig. 7(a). When comparing absolute values of GWED to previous studies it is necessary to keep in mind that GWEDs depend on season, locally different wave sources, and data analysis procedures (e.g., Baumgarten et al., 2017). Nevertheless, studies by Alexander et al. (2011) and Mzé et al. (2014) at Antarctic and mid-latitude stations, respectively, found quantitatively similar results for potential GWEDs averaged over multiple years. Comparing data obtained at high-latitude stations is further affected by the position of the polar vortex, as shown by Whiteway et al. (1997).

Looking at mean GWEDs derived from ECMWF, below $45\,\text{km}$ altitude they are of similar order as the mean total GWEDs derived from lidar data. Above, the mean GWEDs derived from ECMWF are more or less constant with altitude, yielding an underestimation of GWED in ECMWF by factor 3 to 10. This is in line with the underestimated temporal temperature and wind speed variability found in Sect. 4.3.

## 4.5 Larger-scale variations

Applying the method to calculate energy densities not on 1 h profiles (as described in Sect. 3.2) but on all nightly mean temperature and wind speed profiles of one month yields energy densities on a larger time scale. Taking into account only nights with at least 10 h of observations largely reduces the effect of gravity waves and highlights the contribution from planetary

waves or diurnal tides. It has to be noted that applying Eq. (1) to such large-scale variations assumes vertical displacements to be adiabatic and periodic, and advection is neglected. Analogous to the term gravity wave energy density (GWED) we will use the term large-scale wave energy densitiy (LWED) to denote the so derived energy densities. The results for January 2012, January 2014, and January 2015 are shown in Fig. 8, for lidar data (panel a) and ECMWF data (panel b). Compared to GWED, potential and kinetic LWEDs are more variable with altitude and it occurs more often, that potential LWED is larger than

kinetic LWED. Therefore, kinetic-to-potential LWED ratios vary over more than two orders of magnitude. Although total LWEDs show distinct vertical variations, the overall increase with altitude is rather small: It slightly increases in January 2012 (with a local maximum around 70 km altitude) and January 2014 and slightly decreases in January 2015 with a local maximum around 60 km altitude. Contrary to GWED, total LWED derived from ECMWF data is roughly of the same order of magnitude as the total LWED obtained from lidar data, not only in the lower part but in the entire altitude range; e.g., at 61 km

altitude mean total LWEDs range from $\approx 2.2 \cdot 10^2\,\mathrm{J\,kg^{-1}}$ to $\approx 7.3 \cdot 10^2\,\mathrm{J\,kg^{-1}}$ for the lidar data and from $\approx 1.7 \cdot 10^2\,\mathrm{J\,kg^{-1}}$ to $\approx 2.4 \cdot 10^2\,\mathrm{J\,kg^{-1}}$ for the ECMWF data. The kinetic-to-potential energy ratio is larger for the ECMWF data compared to lidar data; especially above 55 km altitude. The explanation is that while the kinetic LWEDs derived from lidar data and ECMWF data are of the same order, the potential LWEDs derived from ECMWF data are smaller than derived from lidar data. Hence, the day-to-day variability of temperatures in ECMWF is too weak, which is visible in Fig. 2 for January 2012.

Comparison of GWED and LWED profiles shows that LWEDs are mainly on the same order of magnitude as GWEDs. Increased mean LWED-to-GWED ratios (up to 10) occur between 60 km and 70 km altitude and below 50 km altitude for potential energy densities, and below 50 km altitude for kinetic energy densities, as is shown in Fig. 9. The total LWED is about 2 to 6 times larger than the total GWED.

## 5 Summary and conclusions

We presented results of more than 300 h of simultaneous temperature and wind observations by Doppler lidar in the Arctic stratosphere and mesosphere, ranging from 30 up to about 85 km altitude, obtained during Januaries 2012, 2014, and 2015.

Considering only these three years, large variability in the mean temperatures and horizontal winds is observed. The temperature and wind data were affected by large-scale dynamics in the middle atmosphere, e.g., an SSW in January 2012. After this minor SSW, two phenomena that are commonly linked to major SSWs (in particular polar vortex split events) were observed

by the ALOMAR RMR lidar: an elevated stratopause and the reformation of the polar vortex. This large-scale activity can be seen for example in the LWED for January 2012 at about 70 km altitude when comparing to altitudes below or the Januaries 2014 and 2015.

We compared mean temperatures and winds from lidar observations to ECMWF and HWM07 data, where we used model data only at times of the lidar observations. Below $\approx 55\,\mathrm{km}$ altitude monthly mean zonal and meridional winds derived from lidar observations and extracted from ECMWF model data agree very well, with differences smaller than $2\,\mathrm{m\,s^{-1}}$ and $5\,\mathrm{m\,s^{-1}}$, respectively. Above, we found differences of up to $20\,\mathrm{K}$, $20\,\mathrm{m\,s^{-1}}$, and $5\,\mathrm{m\,s^{-1}}$ for monthly mean profiles of temperature, zonal, and meridional wind, respectively, between lidar and ECMWF data and of up to $30\,\mathrm{m\,s^{-1}}$ between lidar and HWM07 data.

Analysis of monthly mean gravity wave energy densities showed an increase of total GWED per unit mass with altitude with a scale height of $\approx 16\,\mathrm{km}$, which agrees with previously published values. For one sample night we investigated the ratio of kinetic to potential GWED and found that it varies remarkably with altitude. These variations might be caused by diverse origins of the waves or changing background conditions for wave propagation. Comparison with ECMWF data shows that GWEDs are underestimated in ECMWF by factor 3 to 10 above $50\,\mathrm{km}$ altitude. Analyzing fluctuations of nightly mean profiles allows a similar study for large-scale waves instead of gravity waves. Compared to GWEDs, the LWEDs show larger vertical variations but the overall increase with altitude is smaller. Contrary to GWEDs, the kinetic-to-potential LWED ratios might become smaller 1, this indicates more variability in temperature than in wind, which applies for the remarkable temperature changes in January 2012 at $40\,\mathrm{km}$ and $70\,\mathrm{km}$ altitude in the course of the SSW (cf. Fig. 2). Likewise, a ratio larger 1 indicates larger wind speed variability, e.g., in January 2014 and January 2015 around $50\,\mathrm{km}$ altitude, when the stratopause temperature is quite stable while wind speeds vary strongly (they are affected sensitively by the shape and position of the polar vortex). Total LWEDs derived from ECMWF data agree reasonably well to LWEDs derived from lidar data: E.g., at $61\,\mathrm{km}$ altitude the mean LWEDs derived from lidar and ECMWF data are $\approx 4.5 \cdot 10^2\,\mathrm{J\,kg^{-1}}$ and $\approx 2.0 \cdot 10^2\,\mathrm{J\,kg^{-1}}$, respectively. LWEDs are mainly on the same order of magnitude as GWEDs. Alt altitudes of enhanced large-scale variations, namely between $60\,\mathrm{km}$ and $70\,\mathrm{km}$ altitude for temperatures and below $50\,\mathrm{km}$ altitude for winds, they exceed GWEDs by up to 10. The total LWED is about 2 to 5 times larger than the total GWED.

In future studies daylight data will be included, which will allow to capture tidal effects and extend the analyses to other seasons.

*Acknowledgements.* This study benefited from the excellent support by the dedicated staff at the ALOMAR observatory and the voluntary lidar operators during winter campaigns. The European Centre for Medium-Range Weather Forecasts (ECMWF) is gratefully acknowledged for providing the forecast data; cycles Cy37r3 and Cy40r1 were used in this study. The DoRIS project was supported by Deutsche Forschungs­gemeinschaft (DFG, German Research Foundation, No. BA 2834/1-1). This project has received funding from the European Union's Horizon 2020 Research and Innovation programme under grant agreement No. 653980 (ARISE2), and was supported by the Deutsche Forschungs­gemeinschaft (DFG, German Research Foundation) under project No. LU 1174 (PACOG) and by the German Federal Ministry of Education and Research through the program Role Of The Middle atmosphere In Climate (ROMIC).

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

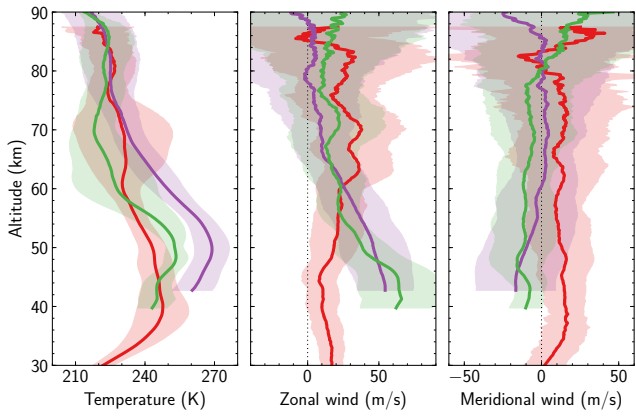

**Figure 1.** January mean temperatures and horizontal winds derived by lidar for the years 2012 (red), 2014 (purple), and 2015 (green). Shaded areas represent the respective standard deviations.

von Zahn, U., von Cossart, G., Fiedler, J., Fricke, K. H., Nelke, G., Baumgarten, G., Rees, D., Hauchecorne, A., and Adolfsen, K.: The ALOMAR Rayleigh/Mie/Raman lidar: objectives, configuration, and performance, Ann. Geophys., 18, 815–833, 2000.

Whiteway, J. A. and Carswell, A. I.: Lidar observations of gravity wave activity in the upper stratosphere over Toronto, J. Geophys. Res., 100, 14,113–14,124, doi:10.1029/95JD00511, 1995.

Whiteway, J. A., Duck, T. J., Donovan, D. P., Bird, J. C., Pal, S. R., and Carswell, A. I.: Measurements of gravity wave activity within and around the Arctic stratospheric vortex, Geophys. Res. Lett., 24, 1387–1390, doi:10.1029/97GL01322, http://dx.doi.org/10.1029/97GL01322, 1997.

Widdel, H.-U.: Vertical movements in the middle atmosphere derived from foil cloud experiments, J. Atmos. Terr. Phys., 49, 723–741, doi:10.1016/0021-9169(87)90015-8, http://www.sciencedirect.com/science/article/pii/0021916987900158, 1987.

Widdel, H.-U.: Foil chaff clouds as a tool for in-situ measurements of atmospheric motions in the middle atmosphere: Their flight behaviour and implications for radar tracking, J. Atmos. Terr. Phys., 52, 89–101, doi:10.1016/0021-9169(90)90071-T, 1990.

Wu, D. L., Schwartz, M. J., Waters, J. W., Limpasuvan, V., Wu, Q., and Killeen, T. L.: Mesospheric Doppler Wind Measurements from Aura Microwave Limb Sounder (MLS), Adv. Space Res., 42, 1246–1252, doi:http://dx.doi.org/10.1016/j.asr.2007.06.014, http://www.sciencedirect.com/science/article/pii/S0273117707006412, 2008.

Xia, H., Dou, X., Sun, D., Shu, Z., Xue, X., Han, Y., Hu, D., Han, Y., and Cheng, T.: Mid-altitude wind measurements with mobile Rayleigh Doppler lidar incorporating system-level optical frequency control method, Opt. Express, 20, 15 286–15 300, doi:10.1364/OE.20.015286, http://www.opticsexpress.org/abstract.cfm?URI=oe-20-14-15286, 2012.

Yuan, T., Thurairajah, B., She, C.-Y., Chandran, A., Collins, R. L., and Krueger, D. A.: Wind and temperature response of midlatitude mesopause region to the 2009 Sudden Stratospheric Warming, J. Geophys. Res., 117, D09 114, doi:10.1029/2011JD017142, http://dx.doi.
org/10.1029/2011JD017142, 2012.

Zink, F. and Vincent, R.: Wavelet analysis of stratosphere gravity wave packets over Macquarie Island 1. Wave parameters, J. Geophys. Res., 2001.

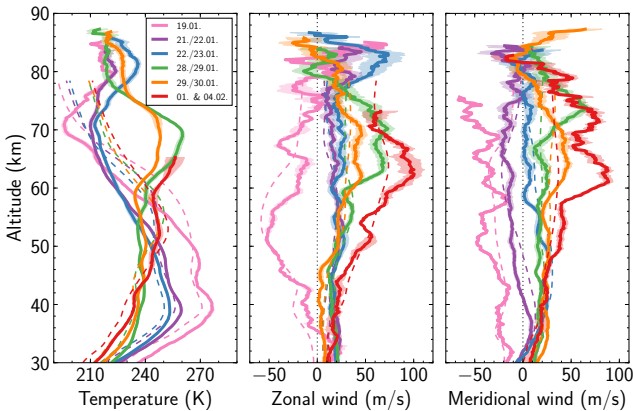

**Figure 2.** Temporal evolution of temperature and horizontal winds during January and early February 2012 after a minor SSW. The profiles are averages of all 1 h profiles of the respective night(s). Solid lines and shaded areas: lidar data and respective standard deviations; dashed lines: ECMWF data with same temporal sampling.

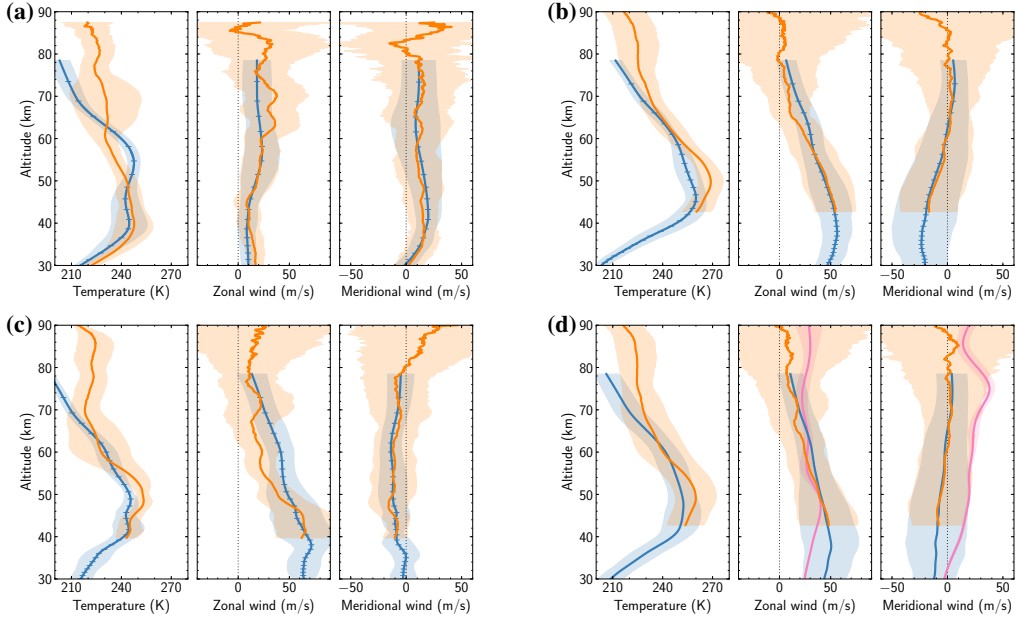

**Figure 3.** January mean temperatures and horizontal winds for the years 2012 (a), 2014 (b), and 2015 (c), and cumulated data (d). ALOMAR RMR lidar (orange), ECMWF (blue), HWM07 (rose). Shaded areas represent the respective standard deviations. The horizontal bars mark the model levels of ECMWF data for one sample profile in each season. The ECMWF cycles used are Cy37r3 for 2012 and Cy40r1 for 2014 and 2015.

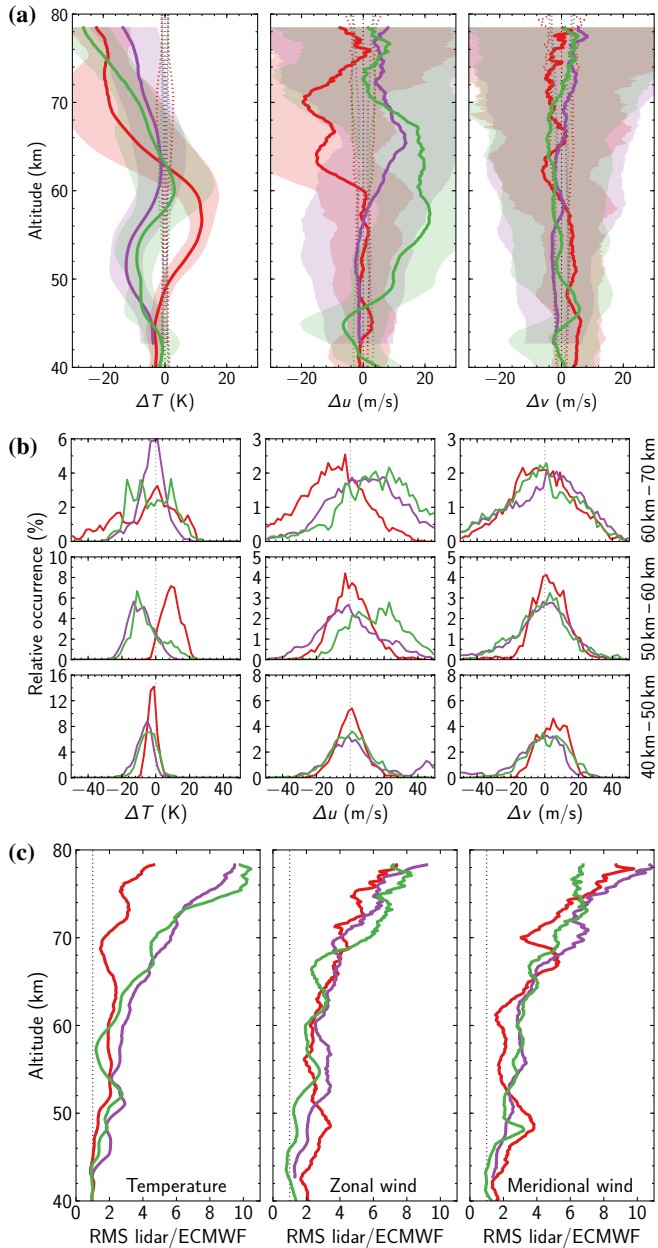

**Figure 4.** Differences between lidar data and ECMWF data for January 2012 (red), January 2014 (purple), and January 2015 (green); the ECMWF cycles used are Cy37r3 for 2012 and Cy40r1 for 2014 and 2015. **(a)** Mean difference $\frac{1}{N}\Sigma(x_{\mathrm{ECMWF}} - x_{\mathrm{lidar}})$; shading represents the respective standard deviations, dotted lines depict the standard error of the mean of the lidar data. **(b)** Distribution of differences $x_{\mathrm{ECMWF}} - x_{\mathrm{lidar}}$ on hourly basis for different altitude ranges. **(c)** Mean ratio of RMS of lidar and ECMWF data. See Tab. 2 for an overview of the number of 1 h profiles taken into account.

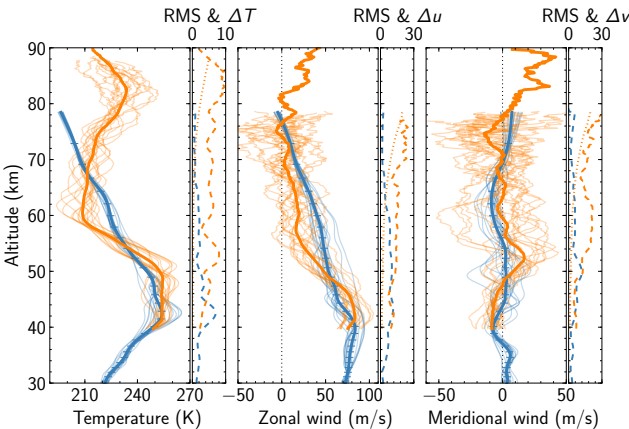

**Figure 5.** Temperature and horizontal winds for the night 20/21 January 2015; lidar (orange), ECMWF (blue). Thin lines denote 1 h profiles, thick lines denote the nightly mean profiles, the horizontal bars mark the model levels of ECMWF data for one sample profile; dashed and dotted lines show the RMS and the mean measurement uncertainty of the 1 h profiles, respectively.

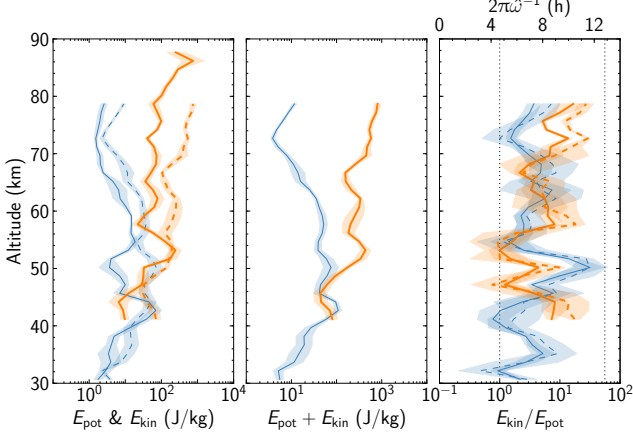

**Figure 6.** Gravity wave energy densities per unit mass and the intrinsic period ($2\pi\hat{\omega}^{-1}$) a monochromatic gravity wave with the given kinetic-to-potential GWED ratio would have, for the night 20/21 January 2015; lidar (orange), ECMWF (blue). Left: potential (solid) and kinetic (dashed) GWED. Middle: total GWED. Right: kinetic-to-potential GWED (solid) and $2\pi\hat{\omega}^{-1}$ (dashed); the dotted vertical lines denote unity and $2\pi f^{-1}$, respectively. Shading represents the respective standard deviation.

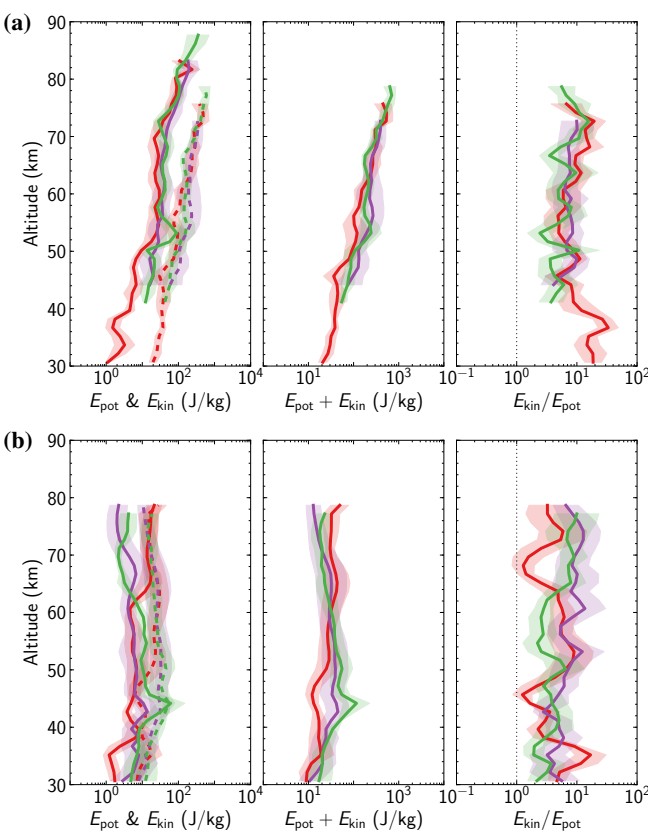

**Figure 7.** January mean gravity wave energy densities for 2012 (red), 2014 (purple), and 2015 (green) derived from lidar data (a) and ECMWF data (b). Shading represents the respective standard deviation. Left: potential (solid) and kinetic (dashed) GWED. Middle: total GWED. Right: kinetic-to-potential GWED.

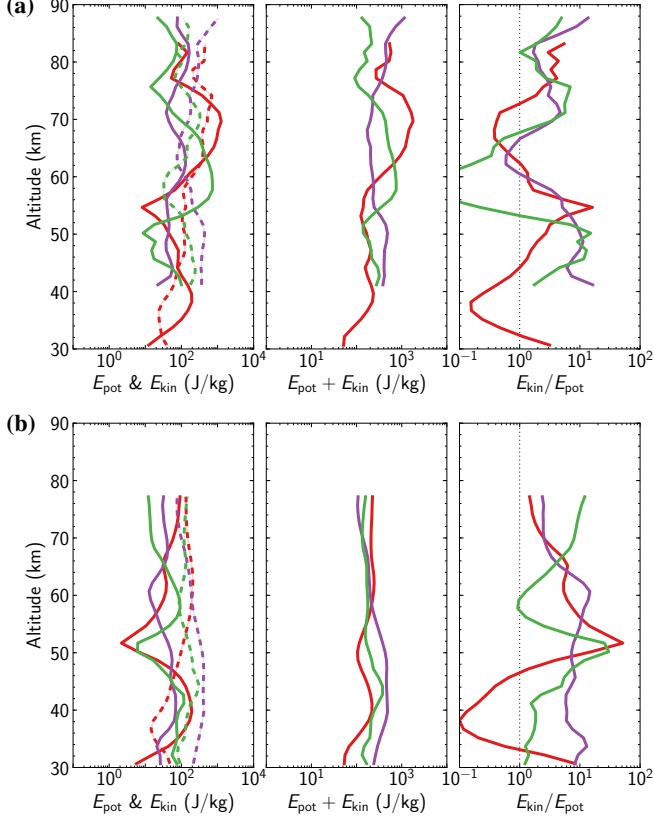

**Figure 8.** January energy densities per unit mass for large-scale waves for 2012 (red), 2014 (purple), and 2015 (green) derived from lidar data (a) and ECMWF data (b); see text for details. Left: potential (solid) and kinetic (dashed) LWEDs. Middle: total LWED. Right: kinetic-to-potential LWED.

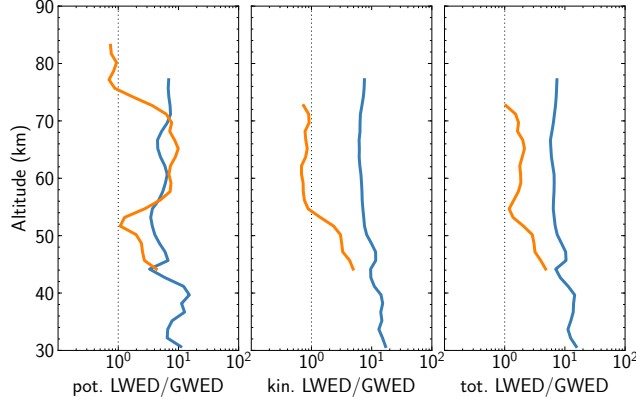

**Figure 9.** Mean LWED-to-GWED ratios for lidar data (orange) and ECMWF data (blue). Left: potential energy densities. Middle: kinetic energy densities. Right: total energy densities.

**Table 1.** List of lidar observations taken into account in this study.

| night | 1 h profiles |
| --- | --- |
| 19/20 January 2012 | 2 |
| 21/22 January 2012 | 15 |
| 22/23 January 2012 | 13 |
| 23/24 January 2012 | 2 |
| 24/25 January 2012 | 3 |
| 28/29 January 2012 | 12 |
| 29/30 January 2012 | 15 |
| 1/2 February 2012 | 1 |
| 3/4 February 2012 | 2 |
| 10/11 January 2014 | 14 |
| 11/12 January 2014 | 17 |
| 14/15 January 2014 | 11 |
| 15/16 January 2014 | 17 |
| 17/18 January 2014 | 11 |
| 18/19 January 2014 | 17 |
| 19/20 January 2014 | 13 |
| 20/21 January 2014 | 11 |
| 21/22 January 2014 | 5 |
| 22/23 January 2014 | 12 |
| 23/24 January 2014 | 1 |
| 24/25 January 2014 | 12 |
| 26/27 January 2014 | 10 |
| 27/28 January 2014 | 5 |
| 29/30 January 2014 | 7 |
| 30/31 January 2014 | 7 |
| 19/20 January 2015 | 16 |
| 20/21 January 2015 | 16 |
| 21/22 January 2015 | 13 |
| 22/23 January 2015 | 16 |
| 23/24 January 2015 | 17 |

**Table 2.** Number of nights and 1 h profiles taken into account for figures showing monthly mean data.

| year | all observations | | long observations ($\geq 10$ h) | |
|---|---|---|---|---|
| | nights | 1 h profiles | nights | 1 h profiles |
| 2012 | 7 | 62 | 4 | 55 |
| 2014 | 16 | 170 | 11 | 145 |
| 2015 | 5 | 78 | 5 | 76[a] |

[a] The observations in the night 21/22 January 2015 consist of two parts of 11 h and 2 h, respectively, separated by a gap of 5 h.