# Peer review of "Winds and temperatures of the Arctic middle atmosphere during January measured by Doppler lidar"

_Atmospheric Chemistry and Physics, 2017_

## Referee Comment (RC1) · Anonymous Referee #2 · 23 Mar 2017

**Review of:**
**Winds and temperatures of the Arctic middle atmosphere during January measured by Doppler lidar**
**by Hildebrand et al.**

The authors present middle atmospheric wind and temperature observations of a lidar system in northern Norway during three Januaries. These observations are compared to the ECMWF and the HWM07 model. Besides the thermal and dynamical mean state, the authors also examine the variability caused by gravity waves and large-scale waves in the observations and the model data.

In a previous review I wrote to the authors "While the collocated middle atmospheric wind and temperature measurements of the Alomar RMR lidar are unique and unprecedented in their temporal and vertical resolution, I find it hard to learn something new from the paper. As it stands right now, the paper is mainly a comparison of different profiles, but no substantial conclusions are drawn from this." This is still the case. Thus, I can only recommend publication of the article after substantial revisions.

Please find my detailed comments below.

**Major comments:**

1) As said before, the paper currently lacks scientific significance. This becomes especially clear when reading the introduction: 50 % of the introduction are a mere review of different techniques to observe wind speeds in the middle atmosphere. The only hint for the importance of wind observations is given in the beginning when the authors state that "together with temperature observations, they [wind observations] also offer more sophisticated studies of gravity waves". Why is this not done in this paper? Showing different profiles of potential and kinetic energy densities does not qualify the paper as a "sophisticated study". To put it short: the paper lacks a scientific question which is investigated and answered in the end. Without a clear scientific question the paper remains unacceptable. A mere publication of the wind and temperature observations is unjustified in my eyes, despite the fact that it is the currently most extensive data set.

2) Most of the very few conclusions drawn by the authors remain rather simple statements which purely describe the observations but the effects which lead to the observations remain in the dark. A few examples:
P. 4, ll. 26–29: the conclusion that the northern hemispheric polar middle atmosphere is highly variable can certainly be considered as textbook knowledge and is therefore redundant.
P. 5, ll. 21–29: the minor SSW and the following elevated stratopause event in 2012 have been well documented by previous studies. Also, as stated correctly by the authors, the mechanism for the formation of an elevated stratopause is known. Hence, I do not see the additional insights which are gained in this study from the combination of wind and temperature observations.
P. 8, l. 33 – p. 9, l. 2: The authors merely speculate on the effects which could cause the different gravity wave propagation conditions. Here, a thorough analysis is needed which investigates the propagation conditions in great detail.
P. 10, l. 9: Why is the Ekin/Epot ratio larger for the ECMWF data compared to the lidar data? What does this imply?

3) P. 8, ll. 25–26: the "approach using energy ratios has the advantage that an (energy weighted) intrinsic period for the ensemble of waves is calculated". This statement is wrong! *Geller and Gong* (2010) derive their formula from the polarization relations which are fulfilled

only for one set of wave parameters $(k, l, m, \hat{\omega})$. If a superposition of waves is to be examined you have to take the sum over the squared wave perturbations in their equations 7) and 8). If you do so and insert the summed polarization relations, you will not end up with a formula, which you can solve for the average frequency. In fact *Geller and Gong* (2010) note in their appendix A1, that their approach always results in larger values of $\hat{\omega}$ than the mean value derived by the hodograph analysis.

Furthermore, it should be noted that according to *Lane et al.* (2003) one can only see long-period inertial gravity waves in the horizontal wind speed fluctuations. Short period gravity waves exhibit more pronounced vertical wind perturbations. Thus the here applied methodology is already biased towards the large period gravity waves.

If the authors want to infer gravity wave periods from their observations they have to use the hodograph approach instead of the energy approach. The energy approach can certainly be taken in the case of a quasi-monochromatic gravity wave field as shown by *Baumgarten et al.* (2015) but for an ensemble of waves it is not applicable.

4) I still think that the comparison of the lidar measurements to the HWM07 model is not appropriate. HWM07 is a climatology and thus one cannot derive a meaningful mean profile from three years of observations in a highly variable surrounding (northern hemispheric polar middle atmosphere) which can be compared to this climatology. As a result the authors cannot differ whether the HWM07 takes too little observations into account (cf. p. 6, ll. 12–13) or whether their observations are simply too few for the comparison. Thus, I recommend removing the paragraph on the HWM07 comparison (p. 6, ll. 6–13) and instead focus the paper more on other aspects.

5) It seems to me that the ECMWF model does not contain any gravity waves above 40–50 km altitude. Here a detailed investigation of the reasons for this behavior is needed. At the moment I do not see any physical reason why the gravity waves should not propagate to higher altitudes than 40–50 km.

6) Regarding the methodology of extracting gravity waves from their observations: The authors state that they do not see any significant differences between their methodology and the Butterworth filter suggested by *Ehard et al.* (2015). If this is not the case, I wonder why the authors do not adopt the Butterworth filter? One of the reasons for using the Butterworth filter is that it ensures a comparability of different studies since the same part of the gravity wave spectrum is extracted from the observations. In fact, *Baumgarten et al.* (2017) recently showed that by applying different methods of gravity wave extraction, a different seasonal cycle of gravity wave activity can be derived.

In a response to my previous review, the authors state that a further reason for not adopting the Butterworth filter is that "When applied to ECMWF data, the Butterworth and the spline method yielded physically dubious results (see Fig. 2): E.g., altitude profiles of GWED derived with the Butterworth method always showed similar oscillating behaviour above $\approx 65$ km altitude; the ratio Ekin=Epot showed values < 1 for the spline and the Butterworth method, which can't be true for gravity waves." This argument can be dismissed in line of my major comment 5), since if there are no gravity waves in the ECMWF model above 40–50 km altitude, the results obtained by all methods are unphysical.

Furthermore, the 10 h averaging applied by the authors has a significant disadvantage when it comes to analyzing the ECMWF data. I guess (see minor comments) that the authors use data from a different ECMWF run after 00 UTC. The corresponding switch from one ECMWF run to another is very likely to introduce a sudden jump of the temperature profile, which will be detected by the authors method, but not by a vertical Butterworth filter. For example the larger Ekin/Epot ratios by the ECMWF compared to the lidar observations (p. 10, l. 9) could

easily be an effect of the different ECMWF runs and analysis used here. In fact I think what you see in the large scale wave energy density is mostly affected by the data assimilation of the ECMWF and not the model dynamics. This has to be investigated with great care!

**Minor comments:**

1) In line with major comment 6): I do not know at which times the authors use analysis data and at which times they use forecast data. For example, ECMWF analysis data is available at 00, 06, 12 and 18 UTC, but one can also retrieve forecast data for these times. Also the authors do not state from which runs the data are taken (i.e. runs initialized at 00 or 12 UTC, or a combination of both). This has to be clarified.
Furthermore, I was wondering, whether you extract the lidar data really at the named position, or whether you interpolate it horizontally to your lidar position?

2) Regarding the measurement uncertainties: At which altitudes do the maximum uncertainties usually appear? How do you treat measurement profiles for which the uncertainties appear at lower altitudes, e.g. 60 km? Do you have further constraints to insure the quality of your observations?

3) P. 5, ll. 12.–13: You state the "also" (why also? what else varies?) small vertical variability of the wind profiles and in the next sentence you state "very pronounced gravity wave structures". Aren't both statements contradictory?

4) P. 5, l. 35: "comparison of lidar data with ECMWF (...) for the whole data set": since you compare two different ECMWF cycles to your observations it is misleading to average both cycles like done in Fig. 4d). In fact it seems to me that by averaging both cycles the deviations between the ECMWF and the observations decrease.
Also on p. 6, l. 19, I am not astonished that the comparison is nonuniform throughout the years, since you compare different cycles to your observations. This has to be evaluated in more detail and with more care!
Also later in ll. 23–26, you should state the cycles used by the other studies.

5) P. 7, l. 4: what is the RMS, I guess the authors mean "root mean square" but of what? Please clarify and also explain the abbreviation. Maybe also give a short explanation as to why an increase of the RMS is "expected for the effect of gravity waves".

6) Figure 4b) is unnecessary and should be removed. The information on the deviation of the different profiles from one another is already contained in the profiles and the according standard deviations (shaded area) in Figure 4a).

7) In my eyes also Figure 5 is unnecessary, since the information on gravity wave activity is already contained in Figure 6 and the paragraph (p. 6, l. 30 – p. 7, ll. 2) does not give substantial new information. Furthermore, the conclusions drawn in this paragraph again remain pure speculation.

8) A general comment regarding the Figures: most axis are rather small and difficult to read. E.g. values of the RMS profiles in Figure 5 cannot be inferred. Furthermore, all plots showing Epot and Ekin on a log axis would definitely benefit from a larger aspect ratio so that concrete values can be inferred by the readers more easily. Furthermore, it should be avoided that plotted values are smaller than the axis values (1st panel, Fig. 3c; 3rd panel, Fig. 8a).

**Technical corrections**

P. 1, l. 4 and throughout the text: "month-mean" should read "monthly mean", the same for "night-mean".

P. 2, l. 8: "then" should read "than"

P. 2, l. 9: give the names for the models (ECMWF, HWM07) at the first appearance of the abbreviations in the text

P. 3, ll. 17–19: it might be of help for the reader to slightly change the order of the sentences: "To retrieve winds (...) The temperature retrieval relies (...) The two individually derived temperature profiles (...)" Also cite *Hauchecorne and Chanin* (1980) for the retrieval of your temperature profile.

P. 4, l. 11: the vertical resolution of the two ECMWF model cycles should be stated.

P. 4, l. 12: what is the vertical resolution of the lidar data? On p. 3, l. 27 you state that the lidar data is smoothed with a "window size of 3 km" is this the vertical resolution of the lidar data? Your profiles look way smoother than just one point every 3 km.

P. 4, l. 32: "or even split, *and* warmer air"

P. 5, l. 9: "Only *a* few days later"

P. 5, ll. 10 & 11: "some 20 K colder/warmer" – colloquial, state precise values

P. 5, ll. 11 & 12: "weak east/west/southward" should read "weakly east/west/southward"

P. 6, l. 16: "way too low" – colloquial, state precise values

P. 6, l. 20: "it is good below 60 km altitude", please quantify. "Good" can mean anything.

P. 6, l. 26: "some deviations in the mesosphere", please quantify.

**References**

Baumgarten, G., J. Fiedler, J. Hildebrand, and F.-J. Lübken (2015), Inertia gravity wave in the stratosphere and mesosphere observed by Doppler wind and temperature lidar, *Geophys. Res. Lett.*, *42*, doi:10.1002/2015GL066991.

Baumgarten, K., M. Gerding, and F.-J. Lübken (2017), Seasonal variation of gravity wave parameters using different filter methods with daylight lidar measurements at midlatitudes, *J. Geophys. Res.*, doi:10.1002/2016jd025916.

Ehard, B., B. Kaifler, N. Kaifler, and M. Rapp (2015), Evaluation of methods for gravity wave extraction from middle-atmospheric lidar temperature measurements, *Atmos. Meas. Tech.*, *8*(11), 4645–4655, doi:10.5194/amt−8−4645−2015.

Geller, M., and J. Gong (2010), Gravity wave kinetic, potential, and vertical fluctuation energies as indicators of different frequency gravity waves, *J. Geophys. Res.*, *115*, D11,111, doi:10.1029/2009JD012266.

Hauchecorne, A., and M. Chanin (1980), Density and temperature profiles obtained by lidar between 35 and 70 km, *Geophys. Res. Lett.*, *7*, 565–568, doi:10.1029/GL007i008p00565.

Lane, T. P., M. J. Reeder, and F. M. Guest (2003), Convectively generated gravity waves observed from radiosonde data taken during MCTEX, *Quart. J. Roy. Meteor. Soc.*, *129*(590), 1731–1740, doi:10.1256/qj.02.196.

---

## Referee Comment (RC2) · Anonymous Referee #1 · 26 Apr 2017

The paper presents wind and temperature measurements by lidar technique at the arctic location of Andoya (69°N). The data are from three Januarys in 2012, 2014 and 2015. The measured night time profiles extend form approx. 30km to 85 km altitude with a temporal resolution of 1 hour. Profiles are compared with corresponding ones from ECMWF and HWM07. Significant differences in temperature and wind between the models and the measurements are reported. In a second part of the paper the authors deduce potential and kinetic gravity wave energy densities based on the measured temporal fluctuations of temperatures and winds.

The paper is carefully and clearly written and easy to follow. Figures are clear and document well the results.

[Figure]

It has to be noted, and the authors clearly summarize this in the introduction, that measured wind profiles are very rare and accordingly very few papers present measured data. Further, the number of publications showing datasets over some extended periods are even more scarce. This paper presents extended data for three Januarys and therefore significantly contributes to an area of middle atmospheric research where the data amount is small so far. This is particularly important as in recent years experimental techniques suffer from declining interest and more weight is put on modeling. Data with high quality as presented in this paper are therefore of extreme value for the validation and improvement of models and they merit to be published. This is particularly true for the data discussed in the current paper.

I therefore recommend to publish the paper with some minor modifications or corrections.

In the section about data, page 3, lines 28 etc. it is not clear how the measurement uncertainties are defined. On the one hand they say that typical values are 0.5K and 3m/s for temperature and wind resp. However then it is said that data with uncertainty values roughly ten times higher are also considered. Please clarify why this large range of uncertainties exists and why you take all these data with high uncertainty into consideration.

Section 4 about results shows high variability in temperature and wind from night to night. The January variability particularly in wind significantly depends on where the measurement is taken with respect to the vortex edge. Indeed the authors several times say that the position of the vortex is important but they do never show where it actually is. Unfortunately it is not possible to find out when the measurement was inside or outside of the vortex. I strongly recommend that the authors separate the data set in two, one with profiles from inside and the other one from outside the vortex. Also the comparison with the models might then change. The large differences between model and data might be explained by such an inappropriate comparison. Section 4.2 as well is linked to the polar vortex and the authors say that a reformation of the vortex took

place. Unfortunately again it is not clear how the situation was at Andoya where the observations took place. Please expand this section regarding the vortex.

Technical corrections:

Abstract line 16: The sentence "The total LWED." does not make sense. Something is lost here . page 3, line 25: . . .. was acquired during the nights in January 2012. . ..

page 6, line 12: either use "this discrepancy" or "these discrepancies"

---

## Author Comment (AC1) · 26 Jun 2017

**Reply to acp-2017-167-RC1-supplement, a review of the manuscript ACP-2017-167 "Winds and temperatures of the Arctic middle atmosphere during January measured by Doppler lidar"**

Jens Hildebrand et al.

June 26, 2017

The authors present middle atmospheric wind and temperature observations of a lidar system in northern Norway during three Januaries. These observations are compared to the ECMWF and the HWM07 model. Besides the thermal and dynamical mean state, the authors also examine the variability caused by gravity waves and large-scale waves in the observations and the model data.

In a previous review I wrote to the authors "While the collocated middle atmospheric wind and temperature measurements of the Alomar RMR lidar are unique and unprecedented in their temporal and vertical resolution, I find it hard to learn something new from the paper. As it stands right now, the paper is mainly a comparison of different profiles, but no substantial conclusions are drawn from this." This is still the case. Thus, I can only recommend publication of the article after substantial revisions.

Please find my detailed comments below.

**Major comments**

1. As said before, the paper currently lacks scientific significance. This becomes especially clear when reading the introduction: $50\,\%$ of the introduction are a mere review of different techniques to observe wind speeds in the middle atmosphere. The only hint for the importance of wind observations is given in the beginning when the authors state that "together with temperature observations, they [wind observations] also offer more sophisticated studies of gravity waves". Why is this not done in this paper? Showing different profiles of potential and kinetic energy densities does not qualify the paper as a "sophisticated study". To put it short: the paper lacks a scientific question which is investigated and answered in the end. Without a clear scientific question the paper remains unacceptable. A mere publication of the wind and temperature observations is unjustified in my eyes, despite the fact that it is the currently most extensive data set.

   Following the suggestions of the short comment SC1 by *Dörnbrack* (2017) we included a quantification of the variability of winds and temperatures measured in the Arctic middle atmosphere; observations that have never be done before.

As mentioned earlier (e.g., *Meriwether and Gerrard*, 2004; *Drob et al.*, 2008; *Dörnbrack et al.*, 2017), wind observations in the middle atmosphere are of interest to infer direction and speed of gravity waves, to provide more input data and tests for empirical models like HWM07.

We highlighted this importance in the introduction.

2. Most of the very few conclusions drawn by the authors remain rather simple statements which purely describe the observations but the effects which lead to the observations remain in the dark. A few examples:

   P. 4, ll. 26–29: the conclusion that the northern hemispheric polar middle atmosphere is highly variable can certainly be considered as textbook knowledge and is therefore redundant.

   By quantifying the variability, as suggested by *Dörnbrack* (2017), we now added additional value to the observations and the comparison to model data.

   P. 5, ll. 21–29: the minor SSW and the following elevated stratopause event in 2012 have been well documented by previous studies. Also, as stated correctly by the authors, the mechanism for the formation of an elevated stratopause is known. Hence, I do not see the additional insights which are gained in this study from the combination of wind and temperature observations.

   We are sorry that the reviewer did not see the new insight, so we tried to clarify this in the manuscript. In summary, we clarify that these are the first direct observations of winds and temperatures during an elevated stratopause event in conjunction with the reformation of the polar vortex. As stated in the manuscript, this situation is not well represented in ECMWF data, highlighting the need for observations.

   We now highlighted in the manuscript why we think the data of this event is worth to be published: To quantify that a state-of-the-art weather model is still having some weaknesses in the middle atmosphere and even more observational data that are not assimilated in the model are needed to provide comparisons for model data.

   P. 8, l. 33 – p. 9, l. 2: The authors merely speculate on the effects which could cause the different gravity wave propagation conditions. Here, a thorough analysis is needed which investigates the propagation conditions in great detail.

   We believe that a detailed investigation of propagation conditions will distract from the main messages and is beyond the scope of this paper. We mention two possible explanations for the observed effect of varying gravity wave propagation: 1. multiple origins of gravity waves; 2. changing background conditions. While the second option is clearly visible in Fig. 5 (large temperature gradient and strong wind shear), the first option can not be excluded.

   We now mention in the manuscript that a clear distinction is not possible.

   P. 10, l. 9: Why is the Ekin/Epot ratio larger for the ECMWF data compared to the lidar data? What does this imply?

In general, a larger $E_{\text{kin}}/E_{\text{pot}}$ ratio indicates a larger ratio of wind fluctuations to temperature fluctuations. Inferring from the left panels of Fig. 8, the kinetic energy densities derived from lidar data and ECMWF data are of the same order, while potential energy densities are smaller in ECMWF data compared to lidar data. Hence, the day-to-day variability of temperatures is weaker in ECMWF than in the observations. This is obvious from the nightly mean profiles of January 2012 shown in Fig. 2.

We now mention this conclusion and the reference to Fig. 2 in the manuscript.

3. P. 8, ll. 25–26: the "approach using energy ratios has the advantage that an (energy weighted) intrinsic period for the ensemble of waves is calculated". This statement is wrong! *Geller and Gong* (2010) derive their formula from the polarization relations which are fulfilled only for one set of wave parameters $(k, l, m, \hat{\omega})$. If a superposition of waves is to be examined you have to take the sum over the squared wave perturbations in their equations 7) and 8). If you do so and insert the summed polarization relations, you will not end up with a formula, which you can solve for the average frequency. In fact *Geller and Gong* (2010) note in their appendix A1, that their approach always results in larger values of $\hat{\omega}$ than the mean value derived by the hodograph analysis.

We have now revised this paragraph, clearly mentioning the assumptions made.

N.B., *Geller and Gong* (2010) found smaller values of $\hat{\omega}$ with the energy ratio method than with the hodograph method, not larger.

Furthermore, it should be noted that according to *Lane et al.* (2003) one can only see long-period inertial gravity waves in the horizontal wind speed fluctuations. Short period gravity waves exhibit more pronounced vertical wind perturbations. Thus the here applied methodology is already biased towards the large period gravity waves.

This limitation of the method is now mentioned in the manuscript.

If the authors want to infer gravity wave periods from their observations they have to use the hodograph approach instead of the energy approach. The energy approach can certainly be taken in the case of a quasi-monochromatic gravity wave field as shown by *Baumgarten et al.* (2015) but for an ensemble of waves it is not applicable.

The hodograph method is only applicable to the case of one single gravity wave, not an ensemble of gravity waves (e.g., *Sato*, 1994). In the case of an ensemble of gravity waves it is hard or even impossible to identify the superposition of ellipses in the zonal and meridional wind fluctuations. Therefore the hodograph method cannot be applied to observations not showing a quasi-monochromatic gravity wave field. On the other hand, the energy ratio approach yields results when applied to observations showing a superposition of gravity waves. In this case it has to be noted, that the so derived $2\pi\hat{\omega}^{-1}$ is not the intrinsic period of a certain wave.

We clearly address this issue in the manuscript now.

4. I still think that the comparison of the lidar measurements to the HWM07 model is not appropriate. HWM07 is a climatology and thus one cannot derive a meaningful mean profile from three years of observations in a highly variable surrounding (northern hemispheric polar middle atmosphere) which can be compared to this climatology. As a result the authors cannot differ whether the HWM07 takes too little observations into account (cf. p. 6, ll. 12–13) or whether their observations are simply too few for the comparison. Thus, I recommend removing the paragraph on the HWM07 comparison (p. 6, ll. 6–13) and instead focus the paper more on other aspects.

   We are aware of the limitations that the reviewer list and they have been clearly stated in the manuscript. However, we think that the comparison to HWM07 is valuable for the scientific community as highlighted by the references given in the manuscript.

5. It seems to me that the ECMWF model does not contain any gravity waves above 40–50 km altitude. Here a detailed investigation of the reasons for this behavior is needed. At the moment I do not see any physical reason why the gravity waves should not propagate to higher altitudes than 40–50 km.

   As mentioned by *Dörnbrack* (2017) "the numerical damping applied in the IFS" leads to an underestimation of the variability of winds and temperatures in the ECMWF data. We now mention in the manuscript that damping mechanisms in the ECMWF are the reason for the underestimation of variability, including a reference to *Jablonowski and Williamson* (2011).

   However, a "detailed investigation" of the behaviour of ECMWF regarding the damping of gravity waves is beyond the scope of this study and might be done by experts of the ECMWF model. This manuscripts provides strong hints that gravity waves are not well represented in the ECMWF model at altitudes above 40–50 km, including quantifications of this underestimation.

6. Regarding the methodology of extracting gravity waves from their observations: The authors state that they do not see any significant differences between their methodology and the Butterworth filter suggested by *Ehard et al.* (2015). If this is not the case, I wonder why the authors do not adopt the Butterworth filter? One of the reasons for using the Butterworth filter is that it ensures a comparability of different studies since the same part of the gravity wave spectrum is extracted from the observations. In fact, *Baumgarten et al.* (2017) recently showed that by applying different methods of gravity wave extraction, a different seasonal cycle of gravity wave activity can be derived.

   Numerous approaches to extract fluctuations caused by gravity waves have been applied to lidar data: filters in altitude (e.g., *Ehard et al.*, 2015), filters in time (e.g., *Rauthe et al.*, 2008), filters in both dimensions (e.g., *Baumgarten et al.*, 2015; *Zhao et al.*, 2017), or the variance method used by *Mzé et al.* (2014). Probably all of these methods have their advantages and drawbacks, and it is simply not possible

to take all of them into account in every study about gravity waves. We mentioned the limitations of the approach we used in this study.

Concerning the comparability of different studies, the gravity wave spectrum taken into account depends not only on the applied vertical filtering technique but also on the temporal sampling of the data.

In a response to my previous review, the authors state that a further reason for not adopting the Butterworth filter is that "When applied to ECMWF data, the Butterworth and the spline method yielded physically dubious results (see Fig. 2): E.g., altitude profiles of GWED derived with the Butterworth method always showed similar oscillating behaviour above $\approx 65$ km altitude; the ratio Ekin=Epot showed values $< 1$ for the spline and the Butterworth method, which can't be true for gravity waves." This argument can be dismissed in line of my major comment 5), since if there are no gravity waves in the ECMWF model above 40–50 km altitude, the results obtained by all methods are unphysical.

Given that it cannot be ruled out that ECMWF data might contain some gravity waves above 40–50 km altitude, the approach applied in this study was the only one of the three approaches tested that allowed to quantify the underestimation of GWED in ECMWF data.

Furthermore, the 10 h averaging applied by the authors has a significant disadvantage when it comes to analyzing the ECMWF data. I guess (see minor comments) that the authors use data from a different ECMWF run after 00 UTC. The corresponding switch from one ECMWF run to another is very likely to introduce a sudden jump of the temperature profile, which will be detected by the authors method, but not by a vertical Butterworth filter. For example the larger Ekin/Epot ratios by the ECMWF compared to the lidar observations (p. 10, l. 9) could easily be an effect of the different ECMWF runs and analysis used here. In fact I think what you see in the large scale wave energy density is mostly affected by the data assimilation of the ECMWF and not the model dynamics. This has to be investigated with great care!

As the large-scale energy density relies on nightly mean profiles, we do not think that by using data of two different ECMWF runs per night the results might be corrupted.

**Minor comments**

1. In line with major comment 6): I do not know at which times the authors use analysis data and at which times they use forecast data. For example, ECMWF analysis data is available at 00, 06, 12 and 18 UTC, but one can also retrieve forecast data for these times. Also the authors do not state from which runs the data are taken (i.e. runs initialized at 00 or 12 UTC, or a combination of both). This has to be clarified.

   As already stated in the manuscript, we use forecast data with 1 h time resolution.

We have clarified in the manuscript that we use both runs: the 00 UTC run for data between midnight and noon and the 12 UTC run for data between noon and midnight.

Furthermore, I was wondering, whether you extract the lidar data really at the named position, or whether you interpolate it horizontally to your lidar position?

We extracted the ECMWF data with horizontal resolution of 0.25° and interpolated these data on pressure levels horizontally to the location of ALOMAR.

This is now clarified in the manuscript.

2. Regarding the measurement uncertainties: At which altitudes do the maximum uncertainties usually appear? How do you treat measurement profiles for which the uncertainties appear at lower altitudes, e.g. 60 km? Do you have further constraints to insure the quality of your observations?

The measurement uncertainties increase with altitude, as the amount of received backscattered laser photons decrease with altitude. Hence, highest uncertainties appear generally at the highest altitudes. Profiles reach only as high as the measurement uncertainty is below the thresholds mentioned in Sect. 3. Raw signal profiles (5 min integration) which are obviously disturbed by poor signal quality (e.g., due to clouds) are discarded prior to the 1 h integration and subsequent temperature and wind retrieval. As only very few profiles were affected, we did not add this technical aspect in the revised manuscript.

We expanded the respective paragraph in the manuscript.

3. P. 5, ll. 12.–13: You state the "also" (why also? what else varies?) small vertical variability of the wind profiles and in the next sentence you state "very pronounced gravity wave structures". Aren't both statements contradictory?

We agree that the phrasing was misleading and clarified it.

4. P. 5, l. 35: "comparison of lidar data with ECMWF (. . . ) for the whole data set": since you compare two different ECMWF cycles to your observations it is misleading to average both cycles like done in Fig. 4d). In fact it seems to me that by averaging both cycles the deviations between the ECMWF and the observations decrease.

Since there is no Fig. 4(d) we assume the reviewer is referring to Fig. 3(d). We like to point out that Fig.s 3(a)–(c) and Fig.s 4(a) and (b) clearly show the results separated for the different model cycles. Since this might have gone undetected we have now added the information about the model cycles in the respective figures captions.

Also on p. 6, l. 19, I am not astonished that the comparison is nonuniform throughout the years, since you compare different cycles to your observations. This has to be evaluated in more detail and with more care!

We have carefully separated the data set according to different model cycles and now highlighted this information in the captions of Fig.s 3 and 4.

It is beyond the scope of this manuscript to investigate differences between ECMWF cycles and why ECMWF data might match differently to certain atmospheric conditions.

Also later in ll. 23–26, you should state the cycles used by the other studies.

*Le Pichon et al.* (2015) use ECMWF IFS cycles 38r1 and 38r2; see their Sect. 2.3 for details. *Rüfenacht et al.* (2014) use "ECMWF operational analysis data" of various cycles (*Rüfenacht et al.*, 2016): "36r2 (September to November 2010), 36r4 (November 2010 to May 2011), 37r2 (May to November 2011), 37r3 (November 2011 to June 2012), 38r1 (June 2012 to June 2013), 38r2 (June to November 2013) and 40r1 (November 2013 to February 2015)".

We now note in the manuscript that other studies use different IFS cycles.

5. P. 7, l. 4: what is the RMS, I guess the authors mean "root mean square" but of what? Please clarify and also explain the abbreviation. Maybe also give a short explanation as to why an increase of the RMS is "expected for the effect of gravity waves".

We now included in the manuscript the abbreviation (root mean square) and clarified that we mean the root mean square of the fluctuations as an indicator of gravity wave activity. We also added the explanation of the expected behaviour.

6. Figure 4b) is unnecessary and should be removed. The information on the deviation of the different profiles from one another is already contained in the profiles and the according standard deviations (shaded area) in Figure 4a).

We have considered removing this panel, but since the shape of the distribution cannot be inferred from Fig. 4(a) we decided to keep this panel.

7. In my eyes also Figure 5 is unnecessary, since the information on gravity wave activity is already contained in Figure 6 and the paragraph (p. 6, l. 30 – p. 7, ll. 2) does not give substantial new information. Furthermore, the conclusions drawn in this paragraph again remain pure speculation.

This figure is the only example showing the actual 1 h profiles of lidar and ECMWF data. Furthermore, the discussions of Fig. 4(c) and Fig. 6 build on this figure.

8. A general comment regarding the Figures: most axis are rather small and difficult to read. E.g. values of the RMS profiles in Figure 5 cannot be inferred. Furthermore, all plots showing Epot and Ekin on a log axis would definitely benefit from a larger aspect ratio so that concrete values can be inferred by the readers more easily. Furthermore, it should be avoided that plotted values are smaller than the axis values (1st panel, Fig. 3c; 3rd panel, Fig. 8a).

We increased the font size of the tick labels and axis labels. As the RMS profiles in Fig. 5 are intended to have quality character only, to qualitatively compare

fluctuations and measurement uncertainties, we see no need to enlarge this figure. Concerning clipped profiles in Fig. 3(c) and Fig. 8(a), we used the same axis scaling for the sake of comparison of various figures.

**Technical corrections**

1. P. 1, l. 4 and throughout the text: "month-mean" should read "monthly mean", the same for "night-mean".

   done

2. P. 2, l. 8: "then" should read "than"

   done

3. P. 2, l. 9: give the names for the models (ECMWF, HWM07) at the first appearance of the abbreviations in the text

   done

4. P. 3, ll. 17–19: it might be of help for the reader to slightly change the order of the sentences: "To retrieve winds (. . . ) The temperature retrieval relies (. . . ) The two individually derived temperature profiles (. . . )" Also cite *Hauchecorne and Chanin* (1980) for the retrieval of your temperature profile.

   done

5. P. 4, l. 11: the vertical resolution of the two ECMWF model cycles should be stated.

   The altitude profiles of the ECMWF data already contained small ticks to mark the respective model levels; indicating that the vertical resolution decreases with altitude.

   We now included in the manuscript that cycle Cy37r3 has 91 model levels and Cy40r1 has 137 model levels.

6. P. 4, l. 12: what is the vertical resolution of the lidar data? On p. 3, l. 27 you state that the lidar data is smoothed with a "window size of 3 km" is this the vertical resolution of the lidar data? Your profiles look way smoother than just one point every 3 km.

   The internal range resolution of the lidar instrument is 50 m; the data were gridded to a raster of 150 m vertical resolution. These data were then smoothed with a running box filter with window size of 3 km.

   We clarified this in Sect. 3.

7. P. 4, l. 32: "or even split, *and* warmer air"

   done by using a semicolon instead of a comma

8. P. 5, l. 9: "Only *a* few days later"

   done

9. P. 5, ll. 10 & 11: "some 20 K colder/warmer" – colloquial, state precise values

   done

10. P. 5, ll. 11 & 12: "weak east/west/southward" should read "weakly east/west/southward"

    done

11. P. 6, l. 16: "way too low" – colloquial, state precise values

    done

12. P. 6, l. 20: "it is good below 60 km altitude", please quantify. "Good" can mean anything.

    done

13. P. 6, l. 26: "some deviations in the mesosphere", please quantify.

    done

**References**

Baumgarten, G., J. Fiedler, J. Hildebrand, and F.-J. Lübken, Inertia gravity wave in the stratosphere and mesosphere observed by doppler wind and temperature lidar, *Geophys. Res. Lett.*, *42*(24), 10,929–10,936, doi:10.1002/2015GL066991, 2015GL066991, 2015.

Baumgarten, K., M. Gerding, and F.-J. Lübken, Seasonal variation of gravity wave parameters using different filter methods with daylight lidar measurements at midlatitudes, *J. Geophys. Res.*, accepted, 2017.

Dörnbrack, A., S. Gisinger, and B. Kaifler, On the interpretation of gravity wave measurements by ground-based lidars, *Atmosphere*, *8*(3), doi:10.3390/atmos8030049, 2017.

Dörnbrack, A., Interactive comment on "Winds and temperatures of the Arctic middle atmosphere during January measured by Doppler lidar" by Jens Hildebrand et al., doi:10.5194/acp-2017-167-SC1, 2017.

Drob, D. P., J. T. Emmert, G. Crowley, J. M. Picone, G. G. Shepherd, W. Skinner, P. Hays, R. J. Niciejewski, M. Larsen, C.-Y. She, J. W. Meriwether, G. Hernandez, M. J. Jarvis, D. P. Sipler, C. A. Tepley, M. S. O'Brien, J. R. Bowman, Q. Wu, Y. Murayama, S. Kawamura, I. M. Reid, and R. A. Vincent, An empirical model of the Earth's horizontal wind fields: HWM07, *J. Geophys. Res.*, 2008.

Ehard, B., B. Kaifler, N. Kaifler, and M. Rapp, Evaluation of methods for gravity wave extraction from middle-atmospheric lidar temperature measurements, *Atmos. Meas. Tech.*, *8*(11), 4645–4655, doi:10.5194/amt-8-4645-2015, 2015.

Geller, M. A., and J. Gong, Gravity wave kinetic, potential, and vertical fluctuation energies as indicators of different frequency gravity waves, *J. Geophys. Res.*, *115*(D11), n/a–n/a, doi:10.1029/2009JD012266, 2010.

Hauchecorne, A., and M.-L. Chanin, Density and temperature profiles obtained by lidar between 35 and 70 km, *Geophys. Res. Lett.*, *7*, 565–568, doi:10.1029/GL007i008p00565, 1980.

Jablonowski, C., and D. L. Williamson, *The Pros and Cons of Diffusion, Filters and Fixers in Atmospheric General Circulation Models*, pp. 381–493, Springer Berlin Heidelberg, Berlin, Heidelberg, doi:10.1007/978-3-642-11640-7_13, 2011.

Lane, T. P., M. J. Reeder, and F. M. Guest, Convectively generated gravity waves observed from radiosonde data taken during mctex, *QJRMS*, *129*(590), 1731–1740, doi:10.1256/qj.02.196, 2003.

Le Pichon, A., J. D. Assink, P. Heinrich, E. Blanc, A. Charlton-Perez, C. F. Lee, P. Keckhut, A. Hauchecorne, R. Rüfenacht, N. Kämpfer, D. P. Drob, P. S. M. Smets, L. G. Evers, L. Ceranna, C. Pilger, O. Ross, and C. Claud, Comparison of co-located independent ground-based middle atmospheric wind and temperature measurements with numerical weather prediction models, *J. Geophys. Res.*, *120*(16), 8318–8331, doi:10.1002/2015JD023273, 2015JD023273, 2015.

Meriwether, J. W., and A. J. Gerrard, Mesosphere inversion layers and stratosphere temperature enhancements, *Rev. Geophys.*, *42*(3), RG3003, doi:10.1029/2003RG000133, 2004.

Mzé, N., A. Hauchecorne, P. Keckhut, and M. Thétis, Vertical distribution of gravity wave potential energy from long-term rayleigh lidar data at a northern middle-latitude site, *J. Geophys. Res.*, *119*(21), 12,069–12,083, doi:10.1002/2014JD022035, 2014JD022035, 2014.

Rauthe, M., M. Gerding, and F.-J. Lübken, Seasonal changes in gravity wave activity measured by Lidars at mid-latitudes, *Atmos. Chem. Phys.*, *8*, 6775–6787, 2008.

Rüfenacht, R., A. Murk, N. Kämpfer, P. Eriksson, and S. A. Buehler, Middle-atmospheric zonal and meridional wind profiles from polar, tropical and midlatitudes with the ground-based microwave doppler wind radiometer wira, *Atmos. Meas. Tech.*, *7*(12), 4491–4505, doi:10.5194/amt-7-4491-2014, 2014.

Rüfenacht, R., K. Hocke, and N. Kämpfer, First continuous ground-based observations of long period oscillations in the vertically resolved wind field of the stratosphere and mesosphere, *Atmos. Chem. Phys.*, *16*(8), 4915–4925, doi:10.5194/acp-16-4915-2016, 2016.

Sato, K., A statistical study of the structure, saturation and sources of inertio-gravity waves in the lower stratosphere observed with the mu radar, *J. Atmos. Terr. Phys.*, 1994.

Zhao, J., X. Chu, C. Chen, X. Lu, W. Fong, Z. Yu, R. M. Jones, B. R. Roberts, and A. Dörnbrack, Lidar observations of stratospheric gravity waves from 2011 to 2015 at McMurdo (77.84°S, 166.69°E), Antarctica: Part I. Vertical wavelengths, periods, and frequency and vertical wavenumber spectra, *J. Geophys. Res.*, pp. n/a–n/a, doi: 10.1002/2016JD026368, 2016JD026368, 2017.

---

## Author Comment (AC2) · 26 Jun 2017

June 26, 2017

Reading the paper and the comment by the reviewer I get the impression that the achievement to observe both wind and temperature fields in the middle atmosphere is largely underestimated by the reviewer. For me, the scientific significance of the paper is at least threefold:

1. the clear and detailed documentation of the simultaneous wind and temperature measurements and a QUANTIFICATION of the variability in wind and temperature over a LARGE height region; even if the conclusion the Arctic winter stratosphere/mesosphere is highly variable is "text book" knowledge, the ultimate quantification can turn this statement into a scientifically significant conclusion

   We now included a discussion of the variability of temperatures and winds within single months, including a quantification for different altitudes.

2. the comparison with model profiles which shows a great agreement up to about 45 km altitude (if I would be the author, I would mention this astonishing agreement much more) – just to make it clear: the authors compare INDEPENDENT data, the lidar profiles were not assimilated into the IFS; above this altitude, the numerical damping applied in the IFS is certainly underestimating the variability found in the observations – this could be a little bit more explained; but again it is the quantification of the agreement and disagreement which make the results scientifically relevant

   We now highlighted the good agreement of winds in lidar data and ECMWF data and improved the inter-comparison of both data sets with additional quantification. And we included a short explanation of the damping of gravity waves in the ECMWF model data, including a reference to a detailed overview of various damping approaches used in atmospheric modelling (*Jablonowski and Williamson*, 2011).

3. the exemplary derivation anf presentation that wind observation are a MUST in order to derive intrinsic wave properties; the recent papers by *Zhao et al.* (2017)

and by *Dörnbrack et al.* (2017) point exactly in this direction and I think the present paper is an excellent contribution to push the need for such observations forward

We now highlighted the importance of wind observations in the introduction by including additional references.

Hope to see this work publsihed soon!

**References**

Dörnbrack, A., S. Gisinger, and B. Kaifler, On the interpretation of gravity wave measurements by ground-based lidars, *Atmosphere*, *8*(3), doi:10.3390/atmos8030049, 2017.

Jablonowski, C., and D. L. Williamson, *The Pros and Cons of Diffusion, Filters and Fixers in Atmospheric General Circulation Models*, pp. 381–493, Springer Berlin Heidelberg, Berlin, Heidelberg, doi:10.1007/978-3-642-11640-7_13, 2011.

Zhao, J., X. Chu, C. Chen, X. Lu, W. Fong, Z. Yu, R. M. Jones, B. R. Roberts, and A. Dörnbrack, Lidar observations of stratospheric gravity waves from 2011 to 2015 at McMurdo (77.84° S, 166.69° E), Antarctica: Part I. Vertical wavelengths, periods, and frequency and vertical wavenumber spectra, *J. Geophys. Res.*, pp. n/a–n/a, doi: 10.1002/2016JD026368, 2016JD026368, 2017.

---

## Author Comment (AC3) · 26 Jun 2017

**Reply to acp-2017-167-RC2,**
**a review of the manuscript ACP-2017-167**
**"Winds and temperatures of the Arctic middle atmosphere during January measured by Doppler lidar"**

Jens Hildebrand et al.

June 26, 2017

The paper presents wind and temperature measurements by lidar technique at the arctic location of Andoya (69° N). The data are from three Januarys in 2012, 2014 and 2015. The measured night time profiles extend form approx. 30km to 85 km altitude with a temporal resolution of 1 hour. Profiles are compared with corresponding ones from ECMWF and HWM07. Significant differences in temperature and wind between the models and the measurements are reported. In a second part of the paper the authors deduce potential and kinetic gravity wave energy densities based on the measured temporal fluctuations of temperatures and winds.

The paper is carefully and clearly written and easy to follow. Figures are clear and document well the results.

It has to be noted, and the authors clearly summarize this in the introduction, that measured wind profiles are very rare and accordingly very few papers present measured data. Further, the number of publications showing datasets over some extended periods are even more scarce. This paper presents extended data for three Januarys and therefore significantly contributes to an area of middle atmospheric research where the data amount is small so far. This is particularly important as in recent years experimental techniques suffer from declining interest and more weight is put on modeling. Data with high quality as presented in this paper are therefore of extreme value for the validation and improvement of models and they merit to be published. This is particularly true for the data discussed in the current paper.

I therefore recommend to publish the paper with some minor modifications or corrections.

**Comments**

1. In the section about data, page 3, lines 28 etc. it is not clear how the measurement uncertainties are defined. On the one hand they say that typical values are 0.5K and 3m/s for temperature and wind resp. However then it is said that data with uncertainty values roughly ten times higher are also considered. Please clarify why

this large range of uncertainties exists and why you take all these data with high uncertainty into consideration.

Measurement uncertainties arise from the statistics of the backscattered laser photons detected. As less photons are recorded for higher altitudes (as there are less air molecules), the measurement uncertainty increases with altitude. Therefore, values with a certain range of measurement uncertainties have to be taken into account. As can be seen in Fig. 5 the thresholds mentioned in Sect. 3 are exceeded for 1 h profiles at $\approx 88$ km and $\approx 78$ km altitude for temperatures and winds, respectively, while nightly mean profiles exceed the thresholds at higher altitudes (since more data are taken into account).

We expanded the respective paragraph in the manuscript.

2. Section 4 about results shows high variability in temperature and wind from night to night. The January variability particularly in wind significantly depends on where the measurement is taken with respect to the vortex edge. Indeed the authors several times say that the position of the vortex is important but they do never show where it actually is. Unfortunately it is not possible to find out when the measurement was inside or outside of the vortex. I strongly recommend that the authors separate the data set in two, one with profiles from inside and the other one from outside the vortex. Also the comparison with the models might then change. The large differences between model and data might be explained by such an inappropriate comparison. Section 4.2 as well is linked to the polar vortex and the authors say that a reformation of the vortex took place. Unfortunately again it is not clear how the situation was at Andoya where the observations took place. Please expand this section regarding the vortex.

To get information about the position of the polar vortex relative to ALOMAR, we examine the potential vorticity at a given potential temperature level, as suggested by *Rex et al.* (1998) and applied by, e.g., *Grooß and Müller* (2003): The edge of the polar vortex is defined as potential vorticity of 36 PVU at the 475 K potential temperature ($\Theta$) level. Using ECMWF data we derive the potential vorticity at $\Theta = 475$ K for each 1 h profile of each night (linear interpolation of potential vorticity from model/pressure levels to $\Theta$ levels). A night is then considered as "inside" or "outside" of the polar vortex, if all (or all but one) 1 h profiles have potential vorticity smaller or larger 36 PVU, respectively; during nights with multiple "inside" and "outside" profiles the vortex edge lies above the site. It has to be noted that the polar vortex might bend and twist and therefore the vortex location as defined at 475 K ($\approx 19$ km altitude) may not always represent the situation in the upper strato- and mesosphere.

Figure 1 shows the same data as Fig. 3 of the manuscript but split depending on relative vortex positions. In the cumulated data (panel d) temperatures are higher inside the vortex than outside, according to expectation. This behaviour is not seen in January 2012 with lower "inside" than "outside" temperatures below 50 km altitude and January 2014 with only very small differences between "inside"

and "outside" temperatures. Note that the "vortex edge" profiles are not intermediate profiles between the "inside" and "outside" profiles. Hence, the temporal development of the dynamics (as discussed in Sect. 4.2 for January 2012) seem to surface more dominant than the – somewhat static – distinction between being inside or outside the polar vortex; furthermore, each data subset consists of few nights only. Therefore, and because the lidar-to-ECMWF comparison seems not to differ fundamentally for the separated data sets, we don't discuss all the aspects mentioned in the manuscript for the separated data.

Nevertheless, we expanded Sect. 4.2 about the SSW in January 2012 and mention for each profile in Fig. 2 of the manuscript to which class ("inside", "outside", "vortex edge") it belongs.

**Technical corrections**

1. Abstract line 16: The sentence "The total LWED." does not make sense. Something is lost here . page 3, line 25: ...was acquired during the nights in January 2012...

   done; done

2. page 6, line 12: either use "this discrepancy" or "these discrepancies"

   done

**References**

Grooß, J.-U., and R. Müller, The impact of mid-latitude intrusions into the polar vortex on ozone loss estimates, *Atmos. Chem. Phys.*, *3*(2), 395–402, doi:10.5194/acp-3-395-2003, 2003.

Rex, M., P. von der Gathen, N. R. P. Harris, D. Lucic, B. M. Knudsen, G. O. Braathen, S. J. Reid, H. De Backer, H. Claude, R. Fabian, H. Fast, M. Gil, E. Kyrö, I. S. Mikkelsen, M. Rummukainen, H. G. Smit, J. Stähelin, C. Varotsos, and I. Zaitcev, In situ measurements of stratospheric ozone depletion rates in the arctic winter 1991/1992: A lagrangian approach, *J. Geophys. Res.*, *103*(D5), 5843–5853, doi:10.1029/97JD03127, 1998.

[Figure]

**Figure 1** Like Fig. 3 of the manuscript, but data set split depending on relative vortex positions. January mean temperatures and horizontal winds for the years 2012 (a), 2014 (b), and 2015 (c), and cumulated data (d). Orange: ALOMAR RMR lidar, blue: ECMWF. Solid lines: inside the polar vortex, dashed lines: outside the polar vortex, dotted lines: vortex edge.

---

## Referee Report (RR1)

Review of paper ID ACP-2017-167-SC1

This paper presents solid, well-referenced work, with interesting and potentially important science through both its direct input and modifications it is likely to stimulate on existing models of wave energy content in the middle atmosphere. The authors are clearly careful about their work. This should also motivate others to make similar measurements. I look forward to it being published soon.

That said, the paper can be greatly improved rhetorically, as it is often hard to follow and it contains much clumsy or incorrect English. I will make a number of editorial suggestions.

First, some general comments:
Abstract
You say that "large year-to-year variations of monthly mean temperatures and winds, which in 2012 are caused by a sudden stratospheric warming." That sounds like the SSW is the only player needed to explain this anomaly. Is that what you want to say, or is it one of perhaps a number of contributors, in which you might say "affected" rather than "caused".

You say that the lidar and ECMWF winds show excellent agreement below ~55 km. I wouldn't have said "excellent", as this implies to me that they are always within the measurement error of one another, which does not appear to be the case from the plots. They are certainly more consistently in agreement than above that altitude, but as your text states: there are "differences of up to ... 20 m/s and 5 m/s, and of up to 30 m/s". By the way, this last "of up to 30 m/s" is confusing. Is it 5 m/s or 30 m/s?

1. Introduction
-   In the review of mesosphere wind measurements the authors should also include meteor radar.
-

2. Instrument
None

3. Data
None

4. Results
On page 6, in comparing HWM07 medridional winds with the lidar and ECMWF, you say it is too strong in the entire altitude range. It is also sometimes of the opposite sign. It seems one thing to under/over estimate the magnitude, but something quite different to get the direction wrong. The former may be a result of averaging or mis-parameterizing, but the latter could be having the wrong physics. I wonder if you shouldn't point out something along this line?

Figure 5 supposedly shows individual 1-h integration profiles. My 600 dpi color printer did not reproduce them. Hopefully, the journal will help you to make sure the graphics are visible.

At the top of page 8, it is not clear what you mean by "down to only one tenth". From the following clause, it seems that you are saying that ECMWF reproduces only about 1/10 of the variability observed. But that is not what the sentence says.

When discussing the GWED in figure 6, you state that the GWED slightly decreases between 53 and 67 km. It seems to me that any mean slope along this range is well within the measurement-induced computational uncertainty, and you should probably state that it is as near as you can tell constant in that range.

On page 10, you state that the calculated GWEDs depend on ... data analysis procedures. Is this about averaging and filtering? Because if using a different procedure, however legitimate, gives a different result, that is not very comforting. Can you please elaborate?

5. Summary and Conclusions
None

Editing Comments (suggestions)
Abstract
- next-to-last sentence: "... ECMWF data show similar results as the lidar data."
Suggest: "... ECMWF data show results similar to the lidar data."
- Last sentence: "... GWED and LWED follows that ..."
Suggest: "... GWED and LWED, it follows that ..."

Introduction
Line 6, do not need the "But", the sentence can read "Not only do ..."
Line 11, suggested change "... observations, specifically regarding ..."
Line 20, suggested change "... by the MLS instrument onboard the Aura satellite ..."

Line 7-8 suggested change "... and allow us to study the interannual variability of
temperatures and winds, the temporal evolution on time scales ..."

Line 30 suggested change "... occur around 45 km altitude, while in 2012 the zonal wind is weak
at this height, and the highest zonal wind speeds occur ..."

Line 2 suggested change "... temperature data at 50 km and 70 km altitudes are 6 K and 21 K
..." (BTW, your use of "respectively" is incorrect in this context)
Lines 16-17 suggested change "The ALOMAR RMR lidar took data during the following days and
weeks, i.e. in the aftermath of the SSW."
Line 18 suggested change "Except for the double-stratopause structure ..."
Line 21 suggested change "In contrast, the westward zonal winds are exceptional ... "
Line 23 suggested change "Based on this definition ..."
Line 24 suggested change "... from ECMWF data, we find that ALOMAR is situated ..."
In the following lines, please be consistent with tenses:
line 27 suggested change "... stratopause was ..."
Line 27-28 suggested change "... zonal winds were weakly eastward ... and meridional winds
developed from weakly southward ..."
Lines 32-34 suggested change "For 28-29 and 29-30 January, the temperature maximum around 40 km
vanished and the highest temperatures shifted upward to around 70 k altitude; at roughly the
same altitude where maxima of zonal and meridional winds occurred."
Line 34 and Page 6 line 1:
suggested change "During the beginning of February, the maxima in temperature, zonal wind and
meridional wind intensified and descended further."

Lines 3-4 suggested change "In contrast to this work, those two studies ..."
Line 6 suggested change "Thus, this observation is evidence that elevated stratopause events
..."
Line 10 suggested change "... this is the first time to our knowledge that an elevated
sratopause was together with reformation of the polar vortex have been observed with a direct
temperature and wind measurement technique."
Line 18 suggested change "... particularly for the end of January ..."
Line 19 suggested change "... polar vortex are not captured in ECMWF."
Line 20 suggested change "comparison" to "correlation"
Line 28 suggested change "insufficiently" to "inadequately"
Line 29 suggested change "The zonal wind is too weak in the upper ..., with differences up to
20 ms^-1."

Lines 6-8 suggested change "In 2012 and 2014 it is very good below 60 km with mean differences
of 2 ms^-1 or less, while above 60 km mean differences are around 20 ms^-1 and 15 ms^-2,

respectively. In 2015, mean differences between 10 and 10 ms^-1 occur throughout the altitude range of 45 to 70 km.
Line 9 suggested change "Mean differences are mostly smaller than or around 5 ms^-1, hence on the same order ..."
Line 11 suggested change "differences" to "difference"
Line 13 suggested change "... mesosphere up to 50% from the true wind speeds."
Line 15 suggested change "Figure 4(b) shows distributions of differences in zonal wind between ECMWF and lidar on an hourly basis for different altitude ranges."
Line 16 suggested change "... are broader for higher altitudes ..."
Lines 21-22 suggested change "... corresponding to the temporal and altitude sampling of the lidar. Despite ..."
Line 35 suggested change "... of the ECMWF data are calculated. Then the monthly average ..."

Line 4 suggested change "... Schroeder et al. (2009). From a comparison ..."
Line 10 suggested change "... sufficiently, either regarding January mean profiles or the variability within individual nights, which are underestimated in ECMWF data."
Lines 11-13 suggested change "... wind affects the calculated energy budget of gravity waves, which are the main source of fluctuations on the scale of a few hours. Resulting gravity wave energy densities are discussed in the next section."
Line 15 suggested change "... allows us to perform wave studies ..."
Lines 16-17 suggested change "... potential and kinetic energy. While the former depends on temperature fluctuations the latter ..."

Line 2 suggested change "GWED, mostly by four to five times ..."
Line 23 suggested change "... of the data presented here, short-period gravity waves ... "
Line 33 suggested change "... the remaining altitude ranges may have different causes: 1. ..."

Line 1 suggested change "... due to the strong zonal wind shear at these altitudes, reducing wind speeds from from 80 ms^-1 to 20ms^-1."
Line 2 suggested change "A clear distinction between these possible explanations ..."
Lines 12-14 suggested change "Although the mean total GWED of January 2015 increases nearly throughout the altitude range ... , the increase is slightly steeper below ~55 km altitude than it is above."
Line 21 suggested change "... keep in mind that GWEDs depend on ..."

Figure 5 caption
change "exemplary" to "sample". "Exemplary" means exceptional, and it does not seem that is what you are trying to say.

---

## Author Response (AR2)

**Reply to acp-2017-167-referee-report-1, a review by Anonymous Referee #2 of the manuscript ACP-2017-167 "Winds and temperatures of the Arctic middle atmosphere during January measured by Doppler lidar"**

Jens Hildebrand et al.

September 22, 2017

- In my major comment #1 I asked the authors to formulate a clear scientific question of their paper. As far as I am concerned, this was neither done, nor did the authors argument against my case. It seems to me as if the authors mostly ignored my comment. Furthermore, to me a mere quantification of a variability without a scientific conclusion does not make a full scientific paper. This is a technical note, which could be published for example in a journal such as "Annales Geophysicae" which encourages such formats.

  We are sorry that the reviewer did not find our modifications of the manuscript to be sufficient. We had expanded the introduction to highlight the scientific relevance of the manuscript. The scientific conclusions are mentioned in the respective section of the manuscript. Regarding the scope of ACP, we find the criteria for publication are fulfilled, reviewers and readers of the manuscript in the open discussion phase are supporting that the work is relevant for ACP.

- In reply to my major comment #3, the authors reformulated that their derived $2\pi\omega^{-1}$ is not the intrinsic frequency. If this is not the case, why not name it differently? By naming it $2\pi\omega^{-1}$ you imply that it is a frequency and most of the readers will believe it to be such. In fact in the caption of their Figure 6 the authors also name it as "the intrinsic period $(2\pi\omega^{-1})$ a monochromat gravity wave with the given kinetic-to-potential GWED ratio would have".

  In the manuscript we discuss only energy ratios. In Fig. 6 we show energy ratios. Only on a secondary axis we have added ticks in terms of $2\pi\hat{\omega}^{-1}$ as some readers found it helpful. The assumptions made to add this scale are clearly listed in the manuscript.

- In my major comment #6, I asked the authors to thoroughly investigate the effect of switching from one ECMWF run to another around 00 UTC. The authors replied to this with "we do not think that by using data of two different ECMWF runs per night the results might be corrupted". This is not a thorough investigation!

At least some proof to back the authors statement is necessary here. As it is right now, it is mere speculation.

Comment #6 contained several points that we addressed in the revision of the manuscript.

We try to clarify one of those points that appeared unclear: The reviewer asked for the influence of the switching of ECMWF runs every 12 hours on the $E_{\mathrm{kin}}/E_{\mathrm{pot}}$ ratio on larger scales (refering to page 10, line 9 of the manuscript at that time). As nightly mean profiles are calculated before the variation of these nightly mean profiles is investigated, we are sure that the switching is smoothed out before further processing.

- Furhermore the authors state concerning their methodology that "Probably all of these methods have their advantages and drawbacks, and it is simply not possible to take all of them into account in every study about gravity waves." While I agree with this statement, I still do not understand why the authors then hold on to a method for which it has been shown (Ehard et al., 2015) that it should not be used! The authors even state that "the approach applied in this study was the only one of the three approaches tested that allowed to quantify the underestimation of GWED in ECMWF data". If the nightly-mean method is the only method which yields this result, but it is known that it has major drawbacks, it makes me very suspicious, that what the authors show here is a mere artifact of their methodology.

We tested three different methods of background estimation (namely nightly mean, Butterworth, and two-dimensional spline) with our lidar data and have not found significant differences in the resulting gravity wave energy densities. Therefore, we do not think that the nightly mean method yields basically wrong results. Furthermore, our procedure accommodates some of the drawbacks of the nightly mean method.

We would like to clarify why we choose the nightly mean method:

We applied three different methods of background estimation (namely nightly mean, Butterworth, and two-dimensional spline) to the lidar data and to the ECMWF data for the whole data set.

Regarding the lidar data we have not found significant differences in the resulting gravity wave energy densities, especially not in the energy ratios. This may be a speciality of our data set and might indicate that stationary waves are not dominating in our data.

We have selected the nightly mean method instead of the Butterworth only due to the results when applying the method to the ECMWF data. As the ECMWF data are not on a uniform altitude grid we have to interpolate these to a new grid before applying the Butterworth filter. When comparing the different methods and their altitude dependence we came to the conclusion that the uncertainties in the interpolation lead to erroneous filtered data. Using the nightly mean method the interpolation errors are suppressed.

So in summary: 1. For the lidar data Butterworth and nightly mean method lead to the same results. 2. For ECMWF data a vertical interpolation needed to apply the Butterworth filter leads to erroneous results.

September 22, 2017

This paper presents solid, well-referenced work, with interesting and potentially important science through both its direct input and modifications it is likely to stimulate on existing models of wave energy content in the middle atmosphere. The authors are clearly careful about their work. This should also motivate others to make similar measurements. I look forward to it being published soon. That said, the paper can be greatly improved rhetorically, as it is often hard to follow and it contains much clumsy or incorrect English. I will make a number of editorial suggestions.

**General comments**

- You say that "large year-to-year variations of monthly mean temperatures and winds, which in 2012 are caused by a sudden stratospheric warming." That sounds like the SSW is the only player needed to explain this anomaly. Is that what you want to say, or is it one of perhaps a number of contributors, in which you might say "affected" rather than "caused".

  As we cannot rule out other contributions we changed this as suggested.

- You say that the lidar and ECMWF winds show excellent agreement below $\approx 55\,\mathrm{km}$. I wouldn't have said "excellent", as this implies to me that they are always within the measurement error of one another, which does not appear to be the case from the plots. They are certainly more consistently in agreement than above that altitude, but as your text states: there are "differences of up to ... $20\,\mathrm{m/s}$ and $5\,\mathrm{m/s}$, and of up to $30\,\mathrm{m/s}$". By the way, this last "of up to $30\,\mathrm{m/s}$" is confusing. Is it $5\,\mathrm{m/s}$ or $30\,\mathrm{m/s}$?

  We changed "excellent" to "good".

  The last "of up to $30\,\mathrm{m\,s^{-1}}$" refers to the differences between lidar data and HWM07 data. We rephrased this sentence to make this more clear.

- In the review of mesosphere wind measurements the authors should also include meteor radar.

  Done.

- On page 6, in comparing HWM07 medridional winds with the lidar and ECMWF, you say it is too strong in the entire altitude range. It is also sometimes of the opposite sign. It seems one thing to under/over estimate the magnitude, but something quite different to get the direction wrong. The former may be a result of averaging or mis-parameterizing, but the latter could be having the wrong physics. I wonder if you shouldn't point out something along this line?

  We now explicitly mention this issue.

  As mentioned in the manuscript, HWM07 takes only a limited number of observations into account for this location and altitude range. Unfortunately we found no information to decide if the differing wind direction is caused by sparse sampling or wrong physics.

- Figure 5 supposedly shows individual 1-h integration profiles. My 600 dpi color printer did not reproduce them. Hopefully, the journal will help you to make sure the graphics are visible.

  I now increased the linewidth.

- At the top of page 8, it is not clear what you mean by "down to only one tenth". From the following clause, it seems that you are saying that ECMWF reproduces only about 1/10 of the variability observed. But that is not what the sentence says.

  We now rephrased this sentence.

- When discussing the GWED in figure 6, you state that the GWED slightly decreases between 53 and 67 km. It seems to me that any mean slope along this range is well within the measurement-induced computational uncertainty, and you should probably state that it is as near as you can tell constant in that range.

  We now rephrased this sentence.

- On page 10, you state that the calculated GWEDs depend on ... data analysis procedures. Is this about averaging and filtering? Because if using a different procedure, however legitimate, gives a different result, that is not very comforting. Can you please elaborate?

  Yes, this is also about averaging and filtering, both in time and altitude, but not only. *Ehard et al.* (2015) presented an overview of different filtering methods and compared the results when applied on synthetic and measured data. *Baumgarten et al.* (2017) presented seasonal cycles of GWPED derived using different filtering methods, i.e., filtering in time or altitude.

Without going into details, it is obvious that different methods are sensitive to different parts of the gravity wave spectrum. Similarly, spectral filters in altitude will yield different results than spectral filters in time. And even when applying the same procedure, the results depend on sampling, averaging, and filtering of the data.

We have listed the relevant parameters in the manuscript, so the results are reproducable. We have made sure that we apply the same methodology to the ECMWF data, so the comparison of lidar and ECMWF is robust.

**Editing comments**

- next-to-last sentence: "... ECMWF data show similar results as the lidar data." Suggest: "... ECMWF data show results similar to the lidar data." done

- Last sentence: "... GWED and LWED follows that ..." Suggest: "... GWED and LWED, it follows that ..." done

- page 2, line 6... done

- page 2, line 11... done

- page 2, line 20... done

- page 3, line 7–8... done

- page 4, line 30... done

- page 5, line 2... done

- page 5, lines 16–17... done

- page 5, line 18... done

- page 5, line 21... done

- page 5, line 23... done

- page 5, line 24... done

- page 5, line 27... done

- page 5, lines 27–28... done

- page 5, lines 32–34... done

- page 5, line 34 and page 6, line 1... done

- page 6, lines 3–4... done

- page 6, line 6... done

- page 6, line 10... adopted partially

- page 6, line 18... done

- page 6, line 19... done

- page 6, line 20... We now changed this phrase to "we compare...".

- page 6, line 28... done

- page 6, line 29... done

- page 7, lines 6–8... done

- page 7, line 9... done

- page 7, line 11... done

- page 7, line 13... done

- page 7, line 15... done

- page 7, line 16... done

- page 7, lines 21–22... done

- page 7, line 35... done

- page 8, line 4... done

- page 8, line 10... done

- page 8, lines 11–13... done

- page 8, line 15... done

- page 8, lines 16–17... done

- page 9, line 2... done

- page 9, line 23... done

- page 9, line 33... done

- page 10, line 1... done

- page 10, line 2... done

- page 10, lines 12–14... done

- page 10, line 21... done

- caption of Fig. 5... Done. We changed other occurrences of "exemplary" too.

**References**

[revised manuscript text omitted]